There are amendments to this paper

# A deep learning system accurately classifies primary and metastatic cancers using passenger mutation patterns

Wei Jiao[1,920], Gurnit Atwal [1,2,3,920], Paz Polak [4,5,6,16,920], Rosa Karlic [7], Edwin Cuppen[8,9], PCAWG Tumor Subtypes and Clinical Translation Working Group, Alexandra Danyi[10], Jeroen de Ridder [10], Carla van Herpen[11], Martijn P. Lolkema [12], Neeltje Steeghs[13], Gad Getz [4,5,6,14], Quaid D. Morris [3,15,921], Lincoln D. Stein[1,2,921]* & PCAWG Consortium

In cancer, the primary tumour's organ of origin and histopathology are the strongest determinants of its clinical behaviour, but in 3% of cases a patient presents with a metastatic tumour and no obvious primary. *Here, as part of the ICGC/TCGA Pan-Cancer Analysis of Whole Genomes (PCAWG) Consortium*, we train a deep learning classifier to predict cancer type based on patterns of somatic passenger mutations detected in whole genome sequencing (WGS) of 2606 tumours representing 24 common cancer types produced by the PCAWG Consortium. Our classifier achieves an accuracy of 91% on held-out tumor samples and 88% and 83% respectively on independent primary and metastatic samples, roughly double the accuracy of trained pathologists when presented with a metastatic tumour without knowledge of the primary. Surprisingly, adding information on driver mutations reduced accuracy. Our results have clinical applicability, underscore how patterns of somatic passenger mutations encode the state of the cell of origin, and can inform future strategies to detect the source of circulating tumour DNA.

[1] Computational Biology Program, Ontario Institute for Cancer Research, Toronto, ON, Canada. [2] Department of Molecular Genetics, University of Toronto, Toronto, ON, Canada. [3] Vector Institute, Toronto, ON, Canada. [4] Broad Institute of MIT and Harvard, Cambridge, MA, USA. [5] Harvard Medical School, Boston, MA, USA. [6] Center for Cancer Research, Massachusetts General Hospital, Boston, MA, USA. [7] Bioinformatics Group, Division of Molecular Biology, Department of Biology, Faculty of Science, University of Zagreb, Horvatovac 102a, Zagreb, Croatia. [8] Hartwig Medical Foundation, Science Park 408, Amsterdam, The Netherlands. [9] Center for Molecular Medicine and Oncode Institute, University Medical Center Utrecht, Utrecht, The Netherlands. [10] Center for Molecular Medicine, University Medical Center Utrecht, Utrecht, The Netherlands. [11] Radboud University Medical Center, Nijmegen, The Netherlands. [12] Department of Medical Oncology, Erasmus MC Cancer Institute, University Medical Center Rotterdam, Dr. Molewaterplein 40, 3015 GD Rotterdam, The Netherlands. [13] Department of Medical Oncology, The Netherlands Cancer Institute, Plesmanlaan 121, 1066 CX Amsterdam, The Netherlands. [14] Department of Pathology, Massachusetts General Hospital, Boston, MA, USA. [15] University of Toronto, Toronto, ON, Canada. [16] Present address: Department of Oncological Sciences, Icahn School of Medicine at Mount Sinai, 15 1425 Madison Ave., New York, NY, USA. [920] These authors contributed equally: Wei Jiao, Gurnit Atwal, Paz Polak. [921] These authors jointly supervised: Quaid D. Morris, Lincoln D. Stein. PCAWG Tumor Subtypes and Clinical Translation Working Group authors and their affiliations appear at the end of the paper. PCAWG Consortium members and their affiliations appear online. *email: lincoln.stein@gmail.com

Human cancers are distinguished by their anatomic organ of origin and their histopathology. For example, squamous cell carcinoma originates in the lung and has a histology similar to the normal squamous epithelium that lines bronchi and bronchioles. Together these two criteria, which jointly reflect the tumour's cell of origin, are the single major predictor of the natural history of the disease, including the age at which the tumour manifests, its factors, growth rate, pattern of invasion and metastasis, response to therapy and overall prognosis. Studies have shown that site-directed therapy based on the tumour's cell of origin is more effective than broad-spectrum chemotherapy[1]. However, it is not always straightforward to determine the origin of a metastatic tumour. In the most extreme case, a clinician may be presented with the challenge of determining the source of a poorly differentiated metastatic cancer when multiple imaging studies have failed to identify the primary ('cancer of unknown primary,' CUPS)[2]. In current clinical practice, pathologists use histological criteria assisted by immunohistochemical stains to determine such tumours' histological type and site of origin[3], but this process can be complex and time-consuming, and some tumours are so poorly differentiated that they no longer express the cell-type-specific proteins needed for unambiguous immunohistochemical classification.

Based on recent large-scale exome and genome-sequencing studies, we know that major tumour types present different patterns of somatic mutation[4–7]. For example, ovarian cancers are distinguished by a high rate of genomic rearrangements[8], chronic myelogenous leukaemias (CML) carry a nearly pathognomonic structural variation involving a t(9;22) translocation leading to a BCR–ABL fusion transcript[9], melanomas have high rates of C > T and G > A transition mutations due to UV damage[10] and pancreatic ductal adenocarcinomas have near-universal activating mutations in the KRAS gene[11]. Recent work has pointed to a strong correlation between the regional somatic mutation rate and chromatin accessibility as measured by DNase I sensitivity and histone mark[12], and has suggested that the cell of origin can be inferred from regional mutation counts[13].

The PCAWG Consortium aggregated whole-genome-sequencing data from 2658 cancers across 38 tumour types generated by the ICGC and TCGA projects. These sequencing data were re-analysed with standardised, high-accuracy pipelines to align to the human genome (reference build hs37d5) and identify germline variants and somatically acquired mutations, as described in the PCAWG Network[7].

This paper asks whether we can use machine-learning techniques to accurately determine tumour organ of origin and histology using the patterns of somatic mutation identified by whole-genome DNA sequencing. One motivation of this effort was to demonstrate the feasibility of a next-generation sequencing (NGS)-based diagnostic tool for tumour-type identification. Due to its stability, DNA is particularly easy to recover from fresh and historical tumour samples; furthermore, because mutations accumulate in DNA, they form a historic record of tumour evolution unaffected by the local, metastatic environment. Here we use deep-learning techniques to explore whether a simple DNA-based sequencing and analysis protocol for tumour-type determination would be a useful adjunct to existing histopathological techniques. Unexpectedly, we find that passenger mutation regional distribution and mutation type are sufficient to discriminate among tumour types with a high degree of accuracy, while driver genes and pathways contribute and provide no improvement to the classifier.

## Results

**Training set**. Using the Pan-cancer Analysis of Whole Genomes (PCAWG) data set[7], we built a series of tumour-type classifiers

using individual sequence-based features and combinations of features. The best-performing classifier was validated against an independent set of tumour genomes to determine overall predictive accuracy, and then tested against a series of metastatic tumours from known primaries to determine the accuracy of predicting the primary from a metastasis.

The full PCAWG data set consists of tumours from 2778 donors comprising 34 main histopathological tumour types, uniformly analysed using the same computational pipeline for quality-control filtering, alignment and somatic mutation calling. However, the PCAWG tumour types are unevenly represented, and several have inadequate numbers of specimens to adequately train and test a classifier. We chose a minimum cut-off of 35 donors per tumour type. In a small number of cases, the same donor contributed both primary and metastatic tumour specimens to the PCAWG data set. In these cases, we used only the primary tumour for training and evaluation, except for the case of the small cohort of myeloproliferative neoplasms (Myeloid-MPN; $N = 55$ samples), for which multiple primary samples were available. In this case, we used up to two samples per donor and partitioned the training and testing sets to avoid having the same donor appear more than once in any training/testing set trial. The resulting training set consisted of 2436 tumours spanning 24 major types (Table 1; Supplementary Data 1).

**Classification using single-mutation feature types**. To determine the predictive value of different mutation features, we trained and evaluated a series of tumour-type classifiers based on single categories of feature derived from the tumour mutation profile. For each feature category we developed a random forest (RF) classifier (see the Methods section). Each classifier's input was the mutational feature profile for an individual tumour specimen, and its output was the probability estimate that the specimen belongs to the type under consideration. Each classifier was trained using a randomly selected set of 75% of samples drawn from the corresponding tumour type. To determine the most likely type for a particular tumour sample, we applied its mutational profile to each of the 24 type-specific classifiers, and selected the type whose classifier emitted the highest probability. To evaluate the performance of the system, we applied stratified fourfold cross-validation by training on three-quarters of the data set and testing against each of the other-quarter specimens. We report overall accuracy as well as recall, precision and the F1 score using the average of all four test data sets (see the 'Methods' section for cross-validation methodology and definitions of terms).

We selected a total of seven mutational feature types spanning three major categories (Table 2):

We assessed mutation distribution. The somatic mutation rate in cancers varies considerably from one region of the genome to the next[5]. In whole-genome sequencing, a major covariate of this regional variation in whole-genome sequences is the epigenetic state of the tumour's cell of origin, with 74–86% of the variance in the mutation density being explained by histone marks and other chromatin features related to open versus closed chromatin[6]. This suggests that tumours sharing similar cells of origin will have a similar topological distribution of mutations across the genome. To capture this, we divided the genome into ~3000 1-Mbp bins across the autosomes (excluding sex chromosomes) and created features corresponding to the number of somatic mutations per bin normalised to the total number of somatic mutations. Mutation rate profiles were created independently for somatic substitutions (SNV), indels, somatic copy-number alterations (CNA) and other structural variations (SV). Note that the vast majority of variants, e.g., at least 99% of the SNVs in nearly all

**Table 1 Distribution of tumour types in the PCAWG training and test data sets.**

| Abbreviation | Organ system | Tumour type | Tumour samples |
|---|---|---|---|
| Liver-HCC | Liver | Liver hepatocellular carcinoma | 306 |
| Panc-AdenoCA | Pancreas | Pancreatic adenocarcinoma | 235 |
| Breast-AdenoCA | Breast | Breast adenocarcinoma | 198 |
| Prost-AdenoCA | Prostate gland | Prostate adenocarcinoma | 189 |
| CNS-Medullo | Brain, cranial nerves and spinal cord | Medulloblastoma | 146 |
| Kidney-RCC | Kidney | Renal cell carcinoma (proximal tubules) | 143 |
| Ovary-AdenoCA | Ovary | Ovarian adenocarcinoma | 112 |
| Skin-Melanoma | Skin | Skin-melanoma | 106 |
| Lymph-BNHL | Lymph nodes | Mature B-cell lymphoma | 105 |
| Eso-AdenoCA | Oesophagus | Oesophageal adenocarcinoma | 98 |
| Lymph-CLL | Blood, bone marrow and hematopoietic sysstem | Chronic lymphocytic leukaemia | 95 |
| CNS-PiloAstro | Brain, cranial nerves and spinal cord | Pilocytic astrocytoma | 89 |
| Panc-Endocrine | Pancreas | Pancreatic neuroendocrine tumour | 85 |
| Stomach-AdenoCA | Stomach | Gastric adenocarcinoma | 70 |
| Head-SCC | Gum, floor of mouth and other mouth | Head/neck squamous cell carcinoma | 57 |
| ColoRect-AdenoCA | Large intestine (excluding appendix) | Colorectal adenocarcinoma | 52 |
| Lung-SCC | Lung and bronchus | Lung squamous cell carcinoma | 48 |
| Thy-AdenoCA | Thyroid gland | Thyroid adenocarcinoma | 48 |
| Myeloid-MPN | Blood, bone marrow and hematopoietic system | Myeloproliferative neoplasm | 46 |
| Kidney-ChRCC | Kidney | Renal cell carcinoma (distal tubules) | 45 |
| Bone-Osteosarc | Bones and joints | Sarcoma, bone | 44 |
| CNS-GBM | Brain, cranial nerves and spinal cord | Diffuse glioma | 41 |
| Uterus-AdenoCA | Uterus, nos | Uterine adenocarcinoma | 40 |
| Lung-AdenoCA | Lung and bronchus | Lung adenocarcinoma | 38 |
| | | | **2436** |

**Table 2 WGS feature types used in classifiers.**

| Feature category | Feature type | Feature count | Description |
|---|---|---|---|
| Mutation distribution | SNV-BIN | 2897 | Number of SNVs per 1-Mbp bin, and per chromosome, normalised against the total number of SNVs per sample |
| | CNA-BIN | 2826 | Number of CNAs per 1-Mbp bin |
| | SV-BIN | 2929 | Number of SVs per 1-Mbp bin, and per chromosome, normalised against the total number of SV per sample |
| | INDEL-BIN | 2757 | Number of SNVs per 1-Mbp bin, and per chromosome, normalised against the total number of INDEL per sample |
| Mutation type | MUT-WGS | 150 | Type of single-nucleotide substitution, double- and triple-nucleotide substitution (plus its adjacent nucleotide neighbours) |
| Driver gene/pathway | GEN | 554 | Presence of an impactful mutation in a suspected driver gene |
| | MOD | 1865 | Presence of an impactful mutation in a gene belonging to a suspected driver pathway |

samples, used for this analysis are non-functional passenger mutations. See Campbell[7] and Li[14] for descriptions of point and structural variations in the PCAWG data set.

We also assessed mutation type. The type of the mutation and its nucleotide neighbours, for example G{C > T}C, is an indicator of the exposure history of the cell of origin to extrinsic and endogenous factors that promote mutational processes[15]. This in turn can provide information on the aetiology of the tumour. For example, skin cancers have mutation types strongly correlated with UV light-induced DNA damage. Reasoning that similar tumour types will have similar mutational exposure profiles, we generated a series of features that represented the normalised frequencies of each potential nucleotide change in the context of its 5′ and 3′ neighbours. Like the mutation distribution, the variants that contribute to this feature category are mostly passengers. Readers are referred to Alexandrov[16] for more information on signature analysis in the PCAWG data set.

Finally, we assessed driver genes/pathways. Some tumour types are distinguished by high frequencies of alterations, in particular driver genes and pathways. For example, melanomas have a high

frequency of BRAF gene mutations[17], while pancreatic cancers are distinguished by KRAS mutations[11]. We captured this in two ways: (1) whether a gene is affected by a driver event as determined by the PCAWG Cancer Drivers Working Group[18], and (2) whether there was an impactful coding mutation in any gene belonging to a known or suspected driver pathway (also see Reyna[19] for cancer pathway analysis performed by the PCAWG Pathway and Networks Working Group). We counted driver events affecting protein-coding genes, long non-coding RNAs and micro-RNAs, but did not attempt to account for alterations in cis-regulatory regions. In all we created ~2000 driver pathway-related features describing potential gene and pathway alterations for each tumour.

The accuracy of individual RF classifiers ranged widely across tumour and feature categories, with a median F1 (harmonic mean of recall and precision) of 0.42 and a range from 0.00 to 0.94 (Fig. 1a, b; Supplementary Fig. 1, Supplementary Data 2). Nine tumour types had at least one well-performing classifier that achieved an F1 of 0.80: CNS-GBM, CNS-PiloAstro, Liver-HCC, Lymph-BNHL, Kidney-RCC, Myeloid-MPN, Panc-AdenoCA,

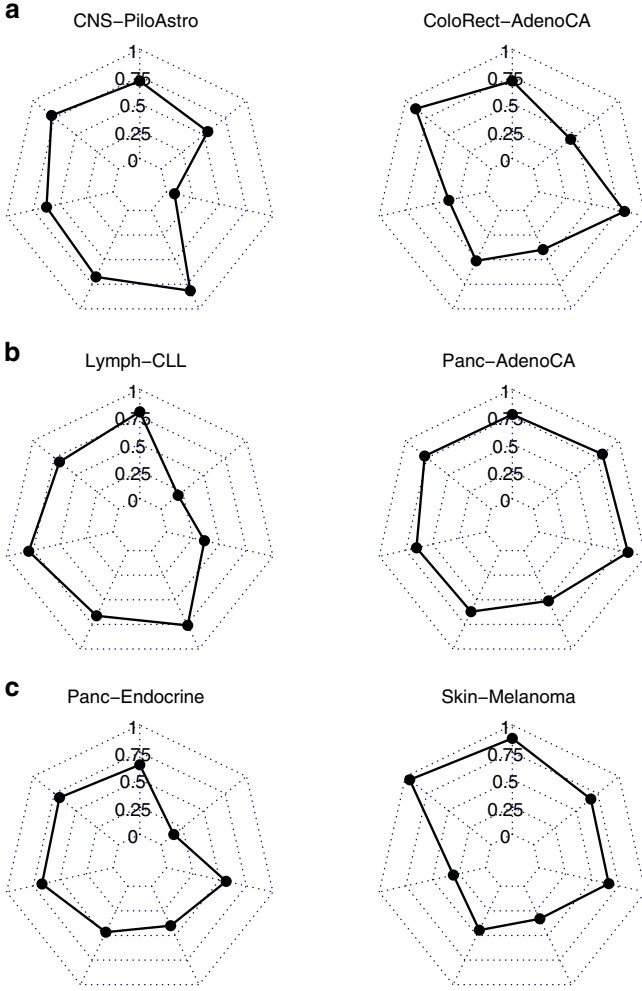

**Fig. 1 Comparison of tumour-type classifiers using single and multiple feature types. a** Radar plots describing the cross-validation-derived accuracy (F1) score of Random Forest classifiers trained on each of 7 individual feature categories, across six representative tumour types. **b** Summary of Random Forest classifier accuracy (F1) trained on individual feature categories across all 24 tumour types. **c** Accuracy of classifiers trained on multiple feature categories. *RF Best Models* corresponds to the cross-validation F1 scores of Random Forest classifiers trained on the three best single-feature categories for all 24 tumour types. *DNN Model* shows the distribution of F1 scores for held-out samples for a multi-class neural network trained using passenger mutation distribution and type. *DNN Model + Drivers* shows F1 scores for the neural net when driver genes and pathways are added to the training features. The centre line in the boxplot represents the median of the F1 scores. The lower and upper bounds of the box represent the first and third quartile. The whiskers extend to 1.5 IQR plus the third quartile or minus the first quantile.

Prost-AdenoCA and Skin-melanoma. Five classifiers performed poorly, with no classifier achieving an accuracy greater than 0.6: Bone-Osteosarc, Head-SCC, Stomach-AdenoCA, Thy-AdenoCA and Uterus-AdenoCA. The remaining eight tumour types had classifiers achieving F1s between 0.60 and 0.80.

The highest accuracies were observed for features related to mutation type and distribution (Fig. 1b). Contrary to our expectations, altered driver genes and pathways were poor discriminatory features. Whereas both SNV type and distribution achieved median F1 scores of ~0.7, RF models built on driver gene or pathway features achieved median F1s of 0.33 and 0.27, respectively. Only Panc-AdenoCA, Kidney-RCC, Lymph-BNHL

and ColoRect-AdenoCA exceeded F1s greater than 0.75 on RF models built from gene or pathway-related features, but we note that even in these cases, the mutation type and/or distribution features performed equally well.

**Classification using combinations of mutation feature types.** We next asked whether we could improve classifier accuracy by combining features from two or more categories. We tested both Random Forest (RF) and multi-class Deep Learning/Neural Network (DNN)-based models (Methods), and found that overall the DNN-based models were more accurate than RF models across a range of feature category combinations (median F1 = 0.86 for RF, F1 = 0.90 for DNN, $p < 1.2e{-}7$ Wilcoxon Rank Sum Test; Fig. 1c). For the DNN-based models, overall accuracy was the highest when just the topological distribution and mutation type of SNVs were taken into account. Adding gene and/or pathway features slightly reduced classification accuracy; using only gene and pathway features greatly reduced classifier performance. We did not investigate the effect of training the DNN on CNV or SV features as these mutation types were not uniformly available in the validation data sets (see below).

Figure 2 shows a heatmap of the DNN classifier accuracy when tested against held-out tumours (mean of 10 independently built models). Overall, the accuracy for the complete set of 24 tumour types was 91% (classification accuracy), but there was considerable variation for individual tumours types (Supplementary Data 3). Recall (also known as sensitivity) ranged from 0.61 (Stomach-AdenoCA) to 0.99 (Kidney-RCC). Precision (similar to specificity but is sensitive to the number of positives in the data set) was comparable, with rates ranging from 0.74 (Stomach-AdenoCA) to 1.00 (CNS-GBM, Skin-Melanoma and Liver-HCC). Twenty-one of 24 tumour types achieved F1s greater than 0.80, including 8 of the 9 types that met this threshold for RF models built on single-feature categories. The three worst-performing tumour types were CNS-PiloAstro (mean F1 0.79 across 10 independently trained DNN models), Lung-AdenoCA (F1 0.77) and Stomach-AdenoCA (F1 0.67).

We investigated the effect of the training set size on classifier accuracy (Fig. 3a). Tumour types with fewer than 100 samples in the data set were more likely to make incorrect predictions, and tumour types with large numbers of samples were among the top performers. However, several tumour types, including ColoRect-AdenoCA ($N = 52$), Lung-SCC ($N = 48$) and CNS-GBM ($N = 41$), achieved excellent predictive accuracy despite having small training sets.

The DNN emits a softmax output that can be interpreted as the probability distribution of the tumour sample across the 24 cancer types. We ordinarily select the highest-probability tumour type as the classifier's choice. If instead we asked how often the correct type is contained among the top N-ranked probabilities, we find that the worst-performing tumour type (Stomach-AdenoCA) achieved a true-positive rate of of 0.88 for placing the correct tumour type among the top ranked three choices, and that the average true-positive rate across all tumour types for this task was 0.98 (Fig. 3b).

**Patterns of misclassification.** Misclassifications produced by the DNN in many cases seem to reflect shared biological characteristics of the tumours. For example, the most frequent classification errors for Stomach-AdenoCA samples were to two other upper gastrointestinal tumours, oesophageal adenocarcinoma (Eso-AdenoCA, 14% misclassification rate), and pancreatic ductal adenocarcinoma (Panc-AdenoCA, 9%). These three organs share a common developmental origin in the embryonic foregut and may share similar epigenetic profiles. We also speculate that the

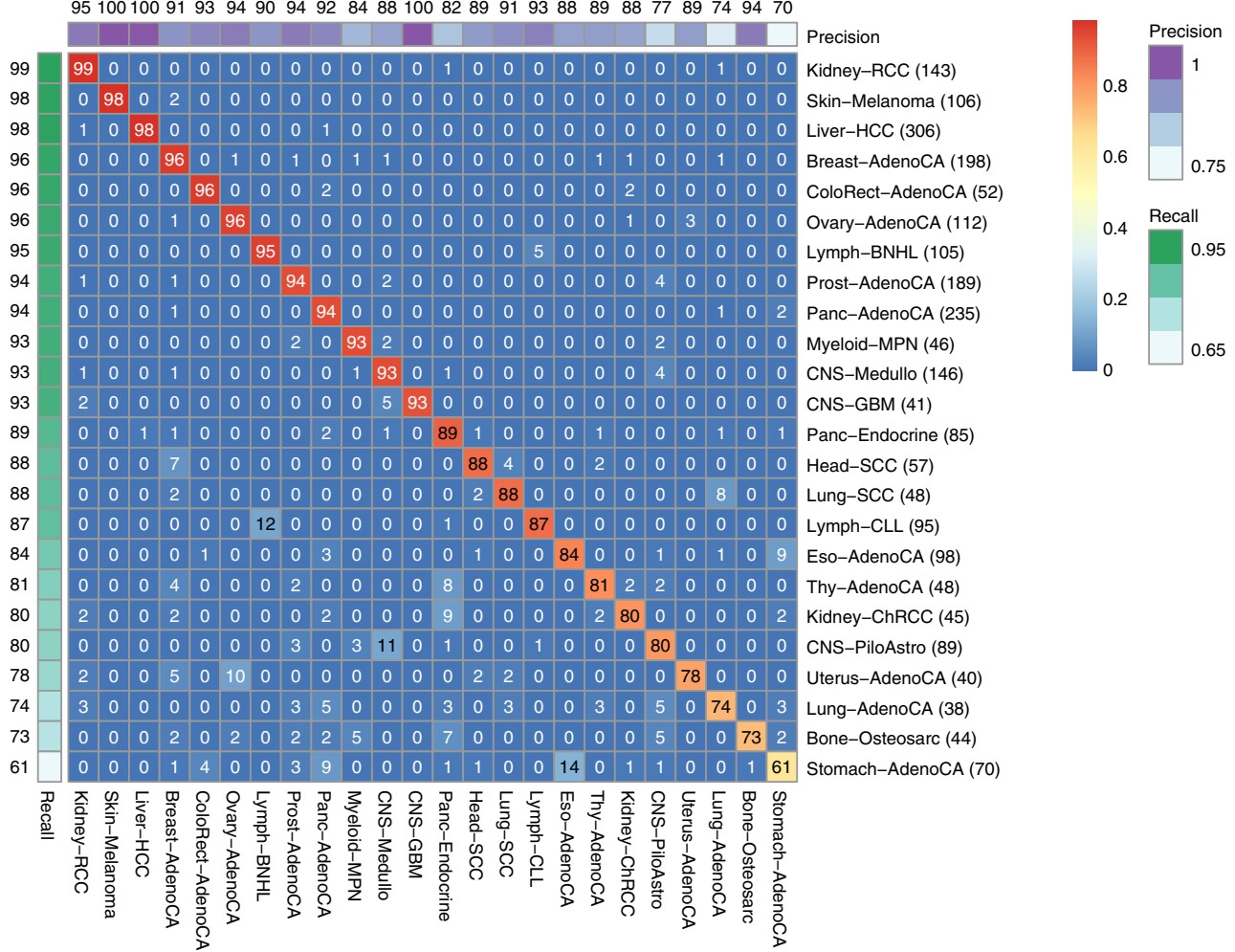

**Fig. 2 Heatmap displaying the accuracy of the merged classifier using a held-out portion of the PCAWG data set for evaluation.** Each row corresponds to the true tumour type; columns correspond to the class predictions emitted by the DNN. Cells are labelled with the percentage of tumours of a particular type that were classified by the DNN as a particular type. The recall and precision of each classifier are shown in the colour bars at the top and left sides of the matrix. All values represent the mean of 10 runs using selected data set partitions. Due to rounding of values, some rows add up to slightly more or less than 100%.

high rate of confusion between gastric and oesophageal cancers might be due to similar mutational exposures among the two sites: a subset of C– > A, C– > G substitutions are commonly seen in stomach and oesophageal (but not pancreatic) cancers and comprise Signature 17 in the COSMIC catalogue of mutational signatures[20]. To test this, we assessed the effect of training the DNN with mutation distribution alone, excluding mutation-type features (Supplementary Fig. 2). Using just passenger mutation distribution, the overall F1 for stomach tumours increased by 4%, supporting the idea that part of the error is due to shared mutational signatures among stomach and oesophageal cancer. Another possible explanation for the frequent misclassification of gastric and oesophageal tumours is that some of the tumours labelled gastric arose at the gastroesophageal junction (GEJ), which some consider to be a distinct subset of oesophageal tumours[21].

Other common misclassification errors include misclassification of 12% of chronic lymphocytic leukaemia (Lymph-CLL) samples as B-cell non-Hodgkin's lymphoma (Lymph-BNHL). Both tumours are derived from the B-cell lymphocyte lineage, and likely share a similar cell of origin. Another pattern was occasional misclassifications among the three types of brain tumour: CNS-GBM, CNS-Medullo and CNS-PiloAstro, all three

of which are derived from various glial lineages. We speculate that these errors are again due to similarities among the cells of origin of these tissues.

Of note is that the DNN was able to accurately distinguish among several tumour types that arise from the same organ. Renal cell carcinoma (Kidney-RCC) and chromophobe renal carcinoma (Kidney-ChRCC) were readily distinguished from each other, as were the squamous and adenocarcinoma forms of non-small-cell lung cancer (Lung-SCC, Lung-AdenoCA), and the exocrine and endocrine forms of pancreatic cancer (Panc-AdenoCA, Panc-Endocrine). The misclassification rate between Lung-SCC and Lung-AdenoCA was just 8%, and all other pairs had misclassification rates of 2% or lower. This is in keeping with a model in which major histological subtypes of tumours reflect different cells of origin.

**Validation on an independent set of primary tumours.** A distinguishing characteristic of the PCAWG data set is its use of a uniform computational pipeline for sequence alignment, quality filtering and variant calling. In real-world settings, however, the data set used to train the classifier may be called using a different set of algorithms than the test data. To assess the

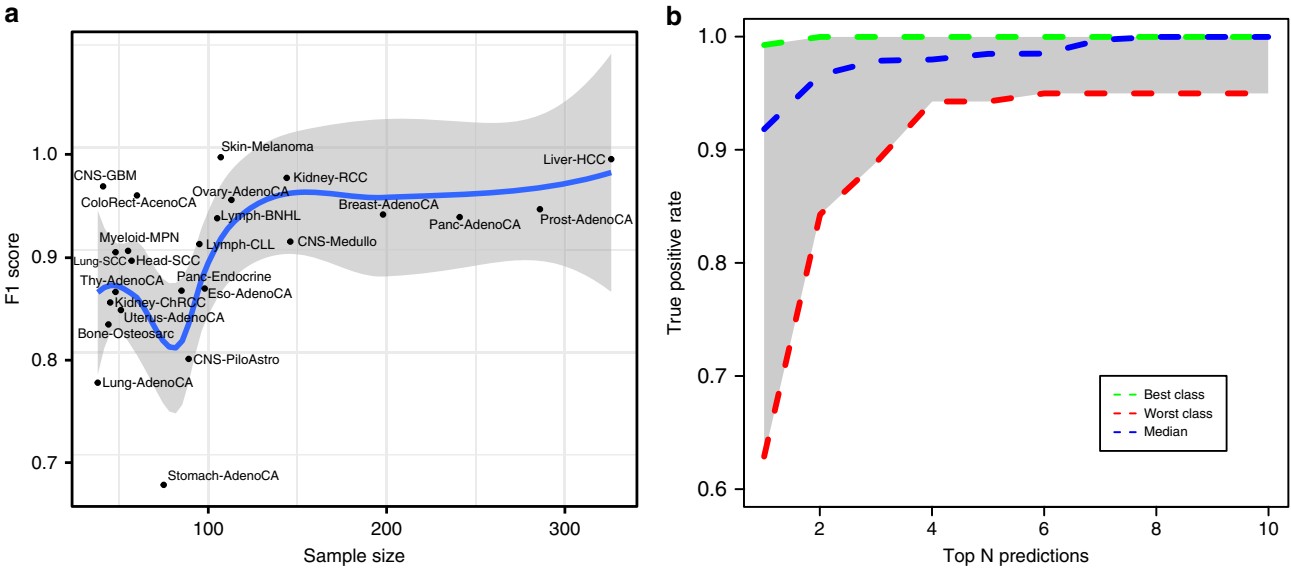

**Fig. 3 Performance of the DNN on held-out PCAWG data. a** The relationship between training set size and prediction accuracy of the DNN is shown for each tumour type. The blue line represents a regression line fit using LOESS regression, while the grey area represents a 95% confidence interval for the regression function. **b** Accuracy of the classifier when it is asked to identify the correct tumour type among its top N-ranked predictions. The blue dashed line is the median true-positive rate among all 24 tumour classes. The green and red dashed lines correspond to the true- positive rate for the best- and worst-performing tumour classes.

accuracy of DNA-based tumour identification when applied in this setting, we applied the classifier trained on PCAWG samples to an independent validation set of 1436 cancer whole genomes assembled from a series of published non-PCAWG projects. The validation set spans 14 distinct tumour types assembled from 21 publications or databases (Supplementary Data 4). We were unable to collect sufficient numbers of independent tumour genomes representing nine of the 24 types in the merged classifier, including colorectal cancer, thyroid adenocarcinoma and lung squamous cell carcinoma. SNV coordinates were lifted from GRCh38 to GRCh37 when necessary, but we did not otherwise process the mutation call sets. With the exception of a set of liver cancer (Liver-HCC) samples in the validation set, which is discussed below, a comparison of the mutation load among each tumour-type cohort revealed no significant differences between the PCAWG and validation data sets (Supplementary Fig. 3).

The DNN classifier recall for the individual tumour types included in the validation data set ranged from 0.41 to 0.98, and the precision ranged from 0.43 to 1.0 (Fig. 4a), achieving an overall accuracy of 88% for classification across the multiple types. In general, the tumour types that performed the best within the PCAWG data set were also the most accurate within the validation, with Breast-AdenoCA, Ovary-AdenoCA, Panc-AdenoCA, Lymph-CLL, CNS-Medullo and Kidney-RCC tumour types all achieving >85% accuracy. The Eso-AdenoCA, Liver-HCC and Paediatric Gliomas were poorly predicted with recalls below 70%, and the remaining types had intermediate accuracies.

The majority of classification errors observed in the primary tumour validation set mirrored the patterns of misclassifications previously observed within the PCAWG samples, with the exception that Liver-HCC cases were frequently misclassified as CNS-Medullo (13%). We believe this case to be due to a lower-than-expected mutation burden in the liver tumours from the validation set (median 3202 SNVs per sample in validation set vs. 22,230 SNVs per sample in the PCAWG training set; $P < 1.5e{-}15$ by Wilcoxon Rank Sum Test; Supplementary Fig. 3). This mutation load is more similar to the rates observed in

CNS-Medullo (median 2330 per sample) among the PCAWG samples, and might suggest poor coverage of Liver-HCC or another sequencing/analysis artefact in the validation set.

We were initially puzzled that a set of 49 validation data set samples that were identified as CNS glioma overwhelmingly matched to the paediatric piloastrocytoma model rather than to the CNS-GBM model. However, on further investigation, we discovered that these samples represent a mixture of low- and high-grade paediatric gliomas, including piloastrocytomas[22–24]. The SNV mutation burden of these paediatric gliomas is also similar to CNS-PiloAstro and significantly lower than adult CNS-GBM (Supplementary Fig. 3).

**Validation on an independent set of metastatic tumours.** To evaluate the ability of the classifier to correctly identify the type of the primary tumour from a metastatic tumour sample, we developed an independent validation data set that combined a published series of 92 metastatic Panc-AdenoCA[25] with an unpublished set of 2,028 metastatic tumours from known primaries across 16 tumour types recently sequenced by the Hartwig Medical Foundation (HMF)[26], resulting in a combined set of 2120 samples across 16 tumour types (Supplementary Data 4). All metastatic samples were subjected to paired-end WGS sequencing of tumour and normal at a tumour coverage of at least $65\times$, but the computational pipelines used for alignment, quality filtering and SNV calling were different from those used for PCAWG. The rules for matching classifier output to the validation set class labels were developed in advance of the experiment, and the DNN classifier was applied to the molecular data from the validation set in a blind fashion.

When the DNN classifier was applied to these metastatic samples, it achieved an overall accuracy of 83% for identifying the type of the known primary (Fig. 4b), which is similar to its performance on the validation primaries. Seven of the tumour types in the metastatic set achieved recall rates of 0.80 or higher, including Breast-AdenoCA (0.97), Kidney (0.96), Panc-AdenoCA (0.94), Prost-AdenoCA (0.86), Skin-Melanoma

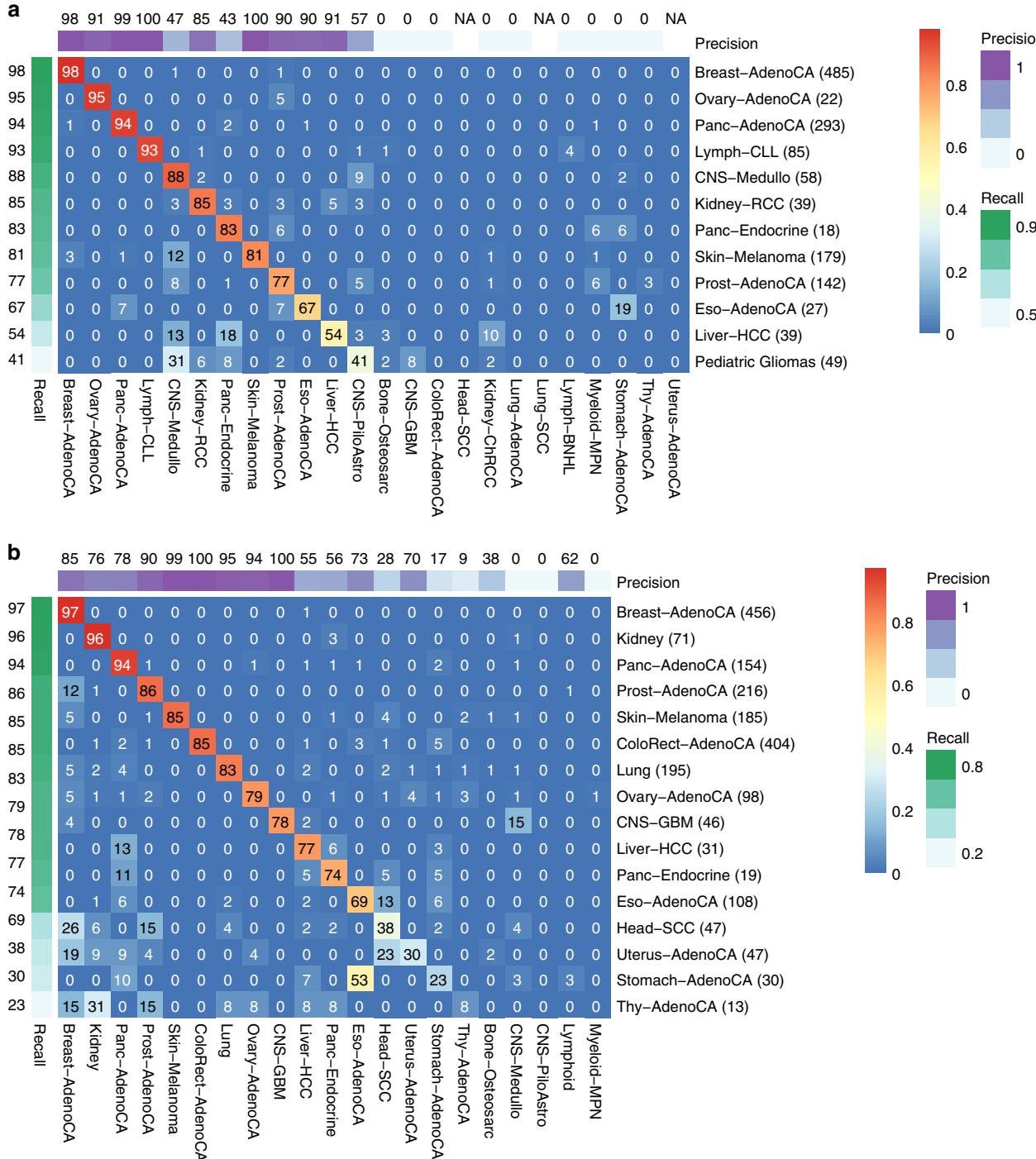

**Fig. 4 Prediction accuracy for the DNN against two independent validation data sets. a** Primary tumours. **b** Metastatic tumours. Each row corresponds to the true tumour type; columns correspond to the class predictions emitted by the DNN. Cells are labelled with the percentage of tumours of a particular type that were classified by the DNN as a particular type. The recall and precision of each classifier are shown in the colour bars at the top and left sides of the matrix. Due to rounding of values, some rows add up to slightly more or less than 100%.

(0.85), ColoRect-AdenoCA (0.85) and Lung (0.83). On the other end of the spectrum, four tumour types failed to achieve a recall of at least 0.50: Head-SCC (0.38), Uterus-AdenoCA (0.30), Stomach-AdenoCA (0.23) and Thyroid-AdenoCA (0.08). Overall, the patterns of misclassification were similar to what was seen within PCAWG. For example, the gastric cancers were misclassified as oesophageal tumours 53% of the time.

In contrast to the other tumour types, metastatic thyroid adenocarcinoma was a clear outlier. In this case, the DNN was unable to correctly identify a great majority of the 13 metastatic samples, classifying them instead as other tumour types such as Kidney, Panc-Endocrine, Prost-AdenoCA or Breast-AdenoCA. We lack information on the histological subtype of the metastatic thyroid tumours in the HMF data set, but speculate that the metastatic thyroid tumours in this set are enriched in more aggressive histological subtypes than the PCAWG primaries, which are exclusively of low-grade papillary ($N = 31$), papillary–follicular ($N = 18$) and papillary–columnar ($N = 1$) types.

The HMF data set also included 62 CUP tumours. While we do not know the corresponding primary for these samples, we did attempt to classify them (Supplementary Data 5). The CUP cases were most frequently classified as Liver-HCC ($N = 10$; 16%), Lung-AdenoCA ($N = 9$; 15%) and Panc-AdenoCA ($N = 8$; 13%). Reassuringly, despite the fact that information on the sex chromosomes was not used by the classifier, almost all the CUP tumours classified as gynaecological tumours (Breast-AdenoCA, $N = 5$; Uterus-AdenoCA, $N = 2$) came from female patients, except one patient with low confident prediction.

## Discussion

Cancer of unknown primary site (CUPS) is a heterogeneous set of cancers diagnosed when a patient presents with metastatic disease, but despite extensive imaging, pathological and molecular studies of the primary cannot be determined[1]. CUPS accounts for 3–5% of cancers, making it the seventh to eighth most frequent type of cancer and the fourth most common cause of cancer death[27]. Even at autopsy, the primary cannot be identified ~70% of the time[28], suggesting regression of the primary in many CUPS cases. CUPS is a clinical dilemma, because therapeutic options are largely driven by tissue of origin, and site-directed therapy is more effective than broad-spectrum chemotherapy[1]. A related diagnostic challenge arises, paradoxically, from the medical community's success in treating cancers and the rising incidence of second primary cancers, now estimated at roughly 16% of incident cancers[29]. Pathologists are often asked to distinguish a late metastatic recurrence of a previously treated primary from a new unrelated primary. However, histopathology alone may be inaccurate at identifying the site of origin of metastases. In one study[30], pathologists who were blinded to the patient's clinical history were able to identify the primary site of a metastasis no more than 49% of the time when given a choice among 11 adenocarcinomas. When asked to rank their guesses, the correct diagnosis was among the top three choices just 76% of the time.

In this paper, we used the largest collection of uniformly processed primary cancer whole genomes assembled to date to develop a supervised machine-learning system capable of accurately distinguishing 24 major tumour types based solely on features that can be derived from DNA sequencing. The accuracy of the system overall when applied in a cross-validation setting was 91%, with 20 of the 24 tumour types achieving an F1 score of 0.83 or higher. When the tumour-type predictions were ranked according to their probability scores, the correct prediction was found among the top three rankings 98% of the time. When applied to external validation data sets, the classifier achieved predictive accuracies of 88% and 83%, respectively, for primary and metastatic tumours. The modestly reduced accuracy in the validation sets is likely due to their differing somatic mutation-calling pipelines, which used different quality-control filters, genome builds and SNV callers from the specimens in the training set.

The regional distribution of somatic passenger mutations across the genome was the single most predictive class of feature, followed by the distribution of mutation types. The regional density of somatic mutations is thought to reflect chromatin accessibility to DNA repair complexes, which in turn relates to the epigenetic state of the cancer's cell of origin. The DNN's predictive accuracy is therefore largely driven by a cell-of-origin signal, aided to a lesser extent by signatures of exposure. The observation that the classifier was able to identify the site of origin for metastatic and primary tumours with similar accuracy suggests that the cell of origin and exposure signals are already established in the early cancer (or its precursor cell) and are not masked by subsequent mutations that occur during tumour evolution.

Unexpectedly, the distribution of functional mutations across driver genes and pathways were poor predictors of tumour type in all but a few tumour types. This surprising finding may be explained by the observation that there are relatively few driver events per tumour (mean 4.6 events per tumour[31]), and affect a set of common biological pathways related to the hallmarks of cancer[32]. This finding may also explain the observation that automated prediction of tumour type by exome or gene panel sequencing has so far met with mixed success (see below).

There was considerable variability in the classification accuracy among tumour types. In most cases, tumour types that were frequently confused with each other had biological similarities, such as related tissues or cells of origin. Technical issues that could degrade predictive accuracy include uneven sequencing coverage, low sample purity, inadequate numbers of samples in the training set and tumour-type heterogeneity. A larger collection of tumours with WGS would allow us to improve the classifier accuracy as well as to train the classifier to recognise clinically significant subtypes of tumours.

There are other ways of identifying the site of origin of a tumour. In cases in which the tumour type is uncertain, pathologists frequently apply a series of antibodies to tissue sections to detect tissue-specific antigens via immunohistochemistry (IHC). The drawback of IHC is that it requires manual interpretation, and the decision tree varies according to the differential diagnosis[3]. Furthermore, IHC is known to be confounded by the loss of antigens in poorly differentiated tumours[33]. In principle, tumour differentiation state should not impact the performance of our classifier because it relies on the distribution of passenger mutations, most of which are already established at the time of tumour initiation. Because of the many different grading systems applied across the PCAWG set, a direct test of this notion is difficult, but we are reassured that the independent set of metastases, which frequently represent a higher grade than the primary, performed as well as the external primary tumour validation set.

An alternative to IHC is molecular profiling of tumours using mRNA or miRNA expression, and several commercial systems are now available to identify the tissue of origin using microarray or qRT-PCR assays[28,34,35]. A recent comparative review[34] of five commercial expression-based kits reported overall accuracies between 76 and 89%; the number of tumour types recognised by each system ranges from 6 to 47 with accuracy tending to decrease as the number of discriminated types increases.

Patterns of DNA methylation are also strongly correlated with the tissue of origin. A recent report[36] demonstrated highly accurate classification of more than 70 central nervous system tumour types using a Random Forest classifier trained on methylation array data. Another recent report[37] showed that an immunoprecipitation-based protocol can recover circulating tumour DNA from patient plasma and accurately distinguish among three tumour types (lung, pancreatic and AML) based on methylation patterns.

Previous work in the area of DNA-based tumour-type identification has used targeted gene panel[38] and whole-exome[39–41] sequencing strategies. The targeted gene-based approach described in Tothill[38] is able to discriminate a handful of tumour types that have distinctive driver gene profiles, and can identify known therapeutic response biomarkers, but does not have broader applicability to the problem of tumour typing. In contrast, the whole-exome sequencing approaches were reported by Marquard[41], Chen[39] and Soh[40] and each used machine-learning approaches to discriminate

among 10, 17 and 28 primary sites, respectively, achieving overall accuracies of 69%, 62% and 78%. Interestingly, all three papers demonstrated that classifiers built on multiple feature categories outperformed those built on a single type of feature, consistent with our findings. We demonstrate here that the addition of whole-genome-sequencing data substantially improves discriminative ability over exome-based features. It is also worth noting that Soh[40] was able to achieve good accuracy using SNVs and CNAs spanning just 50 genes, suggesting that it may be possible to retain high classifier accuracy while using mutation ascertainment across a well-chosen set of whole genomic regions.

In practical terms, whole-genome sequencing and analysis of cancers are becoming increasingly cost-effective, and there is an accelerating trend to apply genome sequencing to routine cancer care in order to identify actionable mutations and to test for the presence of predictive biomarkers. An example of the trend is the National Health Service of the United Kingdom, which recently announced a plan to apply WGS routinely to cancer patients[42]. Given the increasing likelihood that many or most cancers will eventually have genomic profiling, it is attractive to consider the possibility of simultaneously deriving the cancer type using an automated computational protocol. This would serve as an adjunct to histopathological diagnosis, and could also be used as a quality-control check to flag the occasional misdiagnosis or to find genetically unusual tumours. More forward-looking is the prospect of accurately determining the site of origin of circulating cell-free tumour DNA detected in the plasma using so-called liquid biopsies[43], possibly in conjunction with methylome analysis[36,37]. As genome-sequencing technologies continue to increase in sensitivity and decrease in cost, there are realistic prospects for blood tests to detect early cancers in high-risk individuals[44]. The ability to suggest the site and histological type of tumours detected in this way would be invaluable for informing the subsequent diagnostic workup.

In summary, this is the first study to demonstrate the potential of whole-genome sequencing to distinguish major cancer types on the basis of somatic mutation patterns alone. Future studies will focus on improving the classifier performance by training with larger numbers of samples, subdividing tumour types into major molecular subtypes, adding new feature types and adapting the technique to work with clinical specimens such as those from formalin-fixed, paraffin-embedded biopsies and cytologies.

## Methods

**PCAWG training and testing data set**. All variant call data were downloaded from the ICGC Portal (http://dcc.icgc.org/releases/PCAWG/), and all file names given here are relative to this path. Note that controlled tier access credentials are required from the ICGC and TCGA projects as described in https://docs.icgc.org/pcawg/data/. The consensus Somatic SNV and INDEL files (consensus_snv_indel/final_consensus_snv_indel_passonly_icgc.open.tgz and final_consensus_snv_indel_tcga.controlled.tgz) covers 2778 whitelisted samples from 2583 donors. Consensus SV calls from the PCAWG Structural Variation Working Group were downloaded in VCF format (consensus_sv/final_consensus_sv_vcfs_passonly.icgc.controlled.tgz and final_consensus_sv_vcfs_passonly.tcga.controlled.tgz). Ploidy and purity information are from the PCAWG Evolution and Heterogeneity Working Group (consensus_cnv/consensus.20170217.purity.ploidy.txt.gz) and driver events were called by the PCAWG Drivers and Functional Interpretation Group (driver_mutations/TableS3_panorama_driver_mutations_ICGC_samples.controlled.tsv.gz and TableS3_panorama_driver_mutations_TCGA_samples.controlled.tsv.gz). Tumour histological classifications were reviewed and assigned by the PCAWG Pathology and Clinical Correlates Working Group (annotation version 9, August 2016; clinical_and_histology/pcawg_specimen_histology_August2016_v9.xlsx). For model training, we first removed all samples that had been flagged as exhibiting microsatellite instability (MSI) by the PCAWG Technical Working Group (msi/MS_analysis.PCAWG_release_v1.RIKEN.xlsx). In a small number of cases, the same donor contributed both primary and metastatic tumour specimens to the PCAWG data set. In these cases, we used only the primary

tumour for training and evaluation, except for the case of the small cohort of myeloproliferative neoplasms (Myeloid-MPN; $N = 55$ samples), for which multiple primary samples were available. In this case, we used up to two samples per donor and partitioned the training and testing sets to avoid having the same donor appear more than once in any training/testing set trial (see Supplementary Data 1 for the complete list of tumour specimens).

**Independent validation data set: primary and metastatic tumours**. To independently validate the neural network-based classifier, we assembled several sets of tumours that had been subject to whole-genome sequencing outside of PCAWG (Supplementary Data 4).

The primary tumour validation data set consisted of 1236 primary tumours contributed by colleagues participating in the PCAWG Mutational Signatures Working Group and described in ref. [16]. These represent 12 tumour types overlapping with PCAWG types collected from a variety of published studies, non-PCAWG donors submitted to the ICGC data portal (http://dcc.icrg.org) and donors present in the COSMIC database (http://cancer.sanger.ac.uk/cosmic). These independent primaries were supplemented using WGS data from 200 advanced primary pancreatic ductal adenocarcinomas (Panc-AdenoCA) derived from the COMPASS Trial[25] and used with the gracious permission of Dr. Steven Gallinger. In all, the primary tumour validation set contained 1436 primary tumour samples across 12 tumour types. Only tumour types with 10 or more representatives were used for testing.

The metastatic tumour validation data set was derived from SNV calls on 2028 metastatic tumours across 16 tumour types, provided by the Hartwig Medical Foundation (HMF data set). They are a subset of 2090 total samples provided by Dr. Edwin Cuppen with matched PCAWG histology subtypes and are described in Supplementary Data 4 and Priestley[26]. We supplemented this set with 92 metastatic pancreatic ductal adenocarcinomas to the liver from the COMPASS Trial, for a total of 2120 metastatic tumours. As for the primaries, only tumour types with ten or more representatives were tested.

Although the sequencing technologies and genome coverage are comparable among the PCAWG training set and the independent validation data sets, a mixture of different human genome builds, alignment algorithms and SNV calling algorithms were used for the validation data sets. We did not attempt to recall the SNVs, but did lift the genome coordinates of samples that had been aligned to other genome builds to hg19 by CrossMap (Version 0.2.5).

**Human studies approval**. All patients who donated to the PCAWG, COMPASS and HMF data sets consented to international data sharing and secondary analysis of their genomes[25,26,45]. Permission to reanalyse these data was granted by the University of Toronto's Research Ethics Board.

**Somatic mutation feature sets**. Mutational-type features are based on all point substitutions (single-nucleotide variations, SNVs). For each sample, SNVs are categorised across the six possible single-nucleotide changes (A– > C, A– > G, A– > T, C– > A, C– > G and C–T), the 48 possible nucleotide changes plus their 5′ or 3′ flanking base and the 96 possible nucleotide changes plus both flanking nucleotides. This generates 150 mutational-type features in total. The counts in each category are then normalised to the total number of SNVs in the sample.

Mutational distribution features are the number of SNVs, small indels, structural variation (SV) breakpoints and somatic copy-number variations (CNVs) in each 1-megabase bin across the genome. The total number of SNV, indel and SV counts in each bin were normalised to the total number of the corresponding mutational events across the genome. In addition, we generated the following features: (1) the total numbers of each type of mutational event per genome; (2) the number of each type of mutational event per chromosome, normalised by chromosome length; (3) sample purity values; (4) sample ploidy. In total, there are 2897 SNV + indel, 2826 CNV and 2929 SV features. For the initial selection of feature types, we tested all mutational distribution features. However, the final neural network used SNV features only.

Driver gene and pathway features were derived from the driver event list generated by the PCAWG Drivers and Functional Interpretation Working Group[18]. This list contains driver events in coding genes, as well as events that affect miRNA and lncRNAs. We generated a boolean matrix from the list in which each row is a tumour sample and each column is a driver event. To mutations to pathways, we selected any non-synonymous SNV affecting a gene in a pathway, regardless of its putative driver status. These SNVs were then assigned to 1865 pathways from the Reactome resource (http://www.reactome.org, version 58)[46]. A pathway feature was scored as positive if it contained at least one driver gene. Because a gene may be contained within more than one pathway, it is possible for a single driver gene event to generate two or more positive pathway features.

**Machine-learning procedure—Random Forest**. For each of the 24 cancer types selected from the PCAWG sample set, we first used Random Forest model to train classifiers for each cancer type on each of the feature categories described in the above section. The data sets were z-score normalised across the samples before training. We used nested cross-validation to train and test the performance of the

classifiers. In the outer loop, the data set was divided into four folds, and each fold was later used as an independent testing set. In the inner loop, the training portion of the data set was split into three folds, and each fold was used as validation data set to fine-tune the hyperparameters. In the inner loop, we first used a chi-squared test to filter out non-informative (V coefficient equals to 0) features. Then we tuned two hyperparameters for the Random Forest model to achieve the highest cross-validation F1 score. The two hyperparameters were the sample size for positive versus negative classes and the number of trees. We used the default R random-Forest package parameter settings to sample the square root of the number of features at each split of the tree. The code was written in R (version 3.3.0). The main packages used were MLR (version 2.11) and randomForest (4.6–12) in training the model.

**Machine-learning procedure—Neural Network**. We ultimately used a fully connected, feed-forward neural network for the classification of the 24 cancer types based on SNV type and mutational distribution alone. The network had a softmax output, which can be interpreted as a probability distribution of the 24 types. The predicted tumour type was selected by taking the type with the greatest softmax probability.

We used a Bayesian optimisation approach to select hyperparameters[47]. Prior to training, data from PCAWG were split into training, validation and test sets ten times to create ten different partitions over the full data set. For each of the ten partitions, hyperparameters were selected by optimising performance on the validation data for that partition. We used the 'gp_minimize' function from the scikit-optimise 0.5.2 python library[48] to select the following hyperparameters: learning rate for Adam, L2-regularisation penalty (otherwise known as weight decay), dropout rate[49], the number of hidden layers, the number of neurons per hidden layer and activation function. Each model was trained using Adam[50] with a batch size of 32 for 50 epochs. All hyperparameters of Adam other than learning rate were set to the default values specified in the original paper[50]. Bias values were initialised as 0, and all other network weights were initialised using a glorot uniform distribution[51]. The model was evaluated with 200 hyperparameter combinations (i.e., 200 calls to 'gp_minimize' were made). Briefly, 'gp_minimize' approximates a function of model performance based on the hyperparameters with a Guassian Process. For each function call to 'gp_minimize', the performance on the current set of hyperparameters is evaluated by training the neural network, and assessing accuracy on the validation set. Based on this accuracy, the Guassian Process is updated, and a new set of hyperparameters is chosen by optimising an acquisition function. We used expected improvement as the acquisition function. After hyperparameter optimisation, model performance was assessed independently on the corresponding test set for that split. Supplementary Table 1 describes the settings for each of the folds for these hyperparameters.

In order to compare the accuracy of these models with models trained on different feature sets, the procedure above was repeated using driver genes/pathways as input, and again by appending the driver genes/pathway features to the SNV features used above. The final hyperparameter values and model accuracies for each of the trained models are described in Supplementary Data 6. Each model was implemented and trained in Tensorflow 1.10.0[52] and Keras 2.1.5[53]. All code was written in Python 3.6.

**Definitions of accuracy metrics**. To measure the performance of the classifiers, we use the conventional definitions of recall, precision, F1 score and accuracy. In the descriptions below, we use the abbreviations TP (true positive), TN (true negative), FP (false positive) and FN (false negative) to describe correct and incorrect assignments of an unknown tumour to a predicted type, as described by this confusion matrix (Table 3):

*Recall*: The proportion of samples of a particular histopathological type that are correctly assigned to that type:

$$\text{Recall} = \text{TP}/(\text{TP} + \text{FN}). \quad (1)$$

*Precision*: The proportion of samples assigned to a particular type that are truly that type:

$$\text{Precision} = \text{TP}/(\text{TP} + \text{FP}). \quad (2)$$

*F1 Score*: The harmonic mean of recall and precision:

$$\text{F1} = 2(\text{recall} * \text{precision})/(\text{recall} + \text{precision}). \quad (3)$$

*Accuracy*: The proportion of correct assignments. We use this metric only when summarising the performance of the classifier across all 24 tumour types:

$$\text{Accuracy} = (\text{TP} + \text{TN})/(\text{TP} + \text{FP} + \text{TN} + \text{FN}) = \text{correct assignments/total samples.} \quad (4)$$

**Reporting summary**. Further information on research design is available in the Nature Research Reporting Summary linked to this article.

## Data availability
The data sets underpinning the analyses in the paper are detailed in Supplementary Table 2. Aligned sequencing data, as well as somatic and germline variant calls from PCAWG tumours, including single-nucleotide variants, indels, copy-number alterations and structural variants, are available for download at https://dcc.icgc.org/releases/PCAWG. Additional information on accessing the data, including raw read files, can be found at https://docs.icgc.org/pcawg/data/. In accordance with the data access policies of the ICGC and TCGA projects, most molecular, clinical and specimen data are in an open tier that does not require access approval. To access potentially identification information, such as germline alleles and the underlying sequencing data, researchers will need to apply to the TCGA Data Access Committee (DAC) via dbGaP (https://dbgap.ncbi.nlm.nih.gov/aa/wga.cgi?page = login) for access to the TCGA portion of the data set, and to the ICGC Data Access Compliance Office (DACO; http://icgc.org/daco) for the ICGC portion. In addition, to access somatic single-nucleotide variants derived from TCGA donors, researchers will also need to obtain dbGaP authorisation. In addition, the analyses in this paper used a number of data sets that were derived from the raw sequencing data and variant calls (Supplementary Table 2). The individual data sets are available at Synapse (https://www.synapse.org/), and are denoted with *synXXXXX* accession numbers (listed under Synapse ID); all these data sets are also mirrored at https://dcc.icgc.org, with full links, file names, accession numbers and descriptions detailed in Supplementary Table 2. The data sets encompass harmonised tumour histopathology annotations using a standardised hierarchical ontology (syn1038916); driver mutations for each patient from their cancer genome spanning all classes of variants, and coding versus non-coding drivers (syn11639581); clinical data from each patient, including demographics, tumour stage and vital status (syn10389158); inferred purity and ploidy values for each tumour sample (syn8272483). The independent metastatic tumour-independent validation data set generated by the Hartwig Medical Foundation is described in the paper Pan-cancer whole-genome analyses of metastatic solid tumours. Nature. 2019 Oct 23. https://doi.org/10.1038/s41586-019-1689-y. Data are available by application to https://www.hartwigmedicalfoundation.nl/en/applying-for-data/. The remaining metastatic and primary tumour variant call sets used for independent validation have been published and their availability is described in the publications listed in Supplementary Data 4.

## Code availability
The code developed for training and testing the classifier, along with documentation and trained models for the 24 tumour types are available from GitHub at https://github.com/ICGC-TCGA-PanCancer/TumorType-WGS.git. The core computational pipelines used by the PCAWG Consortium for alignment, quality control and variant calling are available to the public at https://dockstore.org/search?search=pcawg under the GNU General Public License v3.0, which allows for reuse and distribution. The code is distributed under the Apache Version 2.0 Open Source license (https://www.apache.org/licenses/LICENSE-2.0).

### Table 3 Confusion matrix.

| Is the unknown sample a member of a particular histopathological type? | Predicted yes | Predicted no |
|---|---|---|
| Actually yes | TP | FN |
| Actually no | FP | TN |

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

## Acknowledgements
We would like to thank Irina Kalatskaya, Quang Trinh, Jared Simpson, Katie Hoadley and David Louis for their helpful comments during preparation of this paper. We also gratefully acknowledge the assistance of Drs. Ludmil B. Alexandrov, Mi Ni Huang, Arnoud Boot, Steven Gallinger, Julie Wilson, Haiko J. Bloemendal, Laurens Beerepoot, Steven G. Rozen and Michael R. Stratton in providing independent WGS primary and metastatic tumour SNV profiles used for validation. We also thank W.J., L.S. and Q.M. supported by funding from the Province of Ontario, Canada. QM's research was supported by a gift from NVIDIA foundation, an advised fund of the Silicon Valley Community Foundation. RK was supported by the European Structural and Investment Funds grant for the Croatian National Centre of Research Excellence in Personalized Healthcare (contract #KK.01.1.1.01.0010), Croatian National Centre of Research Excellence for Data Science and Advanced Cooperative Systems (contract KK.01.1.1.01.0009), the European Commission Seventh Framework Program (Integra-Life; grant 315997) and Croatian Science Foundation (grant IP-2014-09-6400). J.d.R. is supported by a NWO-Vidi grant (016.Vidi.178.023). We acknowledge the contributions of the many clinical networks across ICGC and TCGA who provided samples and data to the PCAWG Consortium, and the contributions of the Technical Working Group and the Germline Working Group of the PCAWG Consortium for collation, realignment and harmonised variant calling of the cancer genomes used in this study. We thank the patients and their families for their participation in the individual ICGC and TCGA projects.

## Author contributions
W.J. performed analyses, wrote and edited the paper. G.A. performed analyses, wrote and edited the paper. E.C. performed analyses and provided data. L.S. conceived the study, and wrote and edited the paper. P.P. conceived the study, performed analyses and edited the paper. G.G. edited the paper and provided discussion. R.K. edited the paper. J.d.R. provided the data and discussion. C.v.H. provided data and edited the paper. M.P.L. provided data. N.S. provided data and edited the paper. A.D. provided data. Q.M. wrote and edited the paper and provided discussion. The PCAWG Tumour Subtypes and Clinical Translation Working Group provided valuable advice and feedback. Andrew V Bianki, Levi Garraway, Sean M Grimmond, Katherine A Hoadley and Lincoln D Stein are working groups or project co-leaders. The ICGC/TCGA PCAWG Network provided data.

## Competing interests
The authors declare no competing interests.

## Additional information

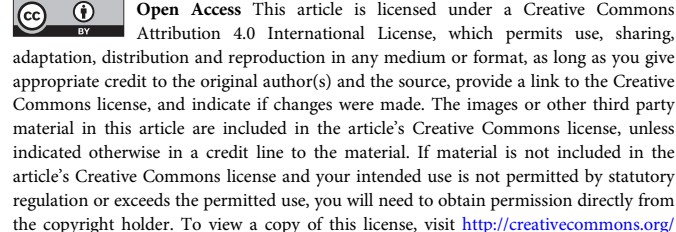

## PCAWG Tumor Subtypes and Clinical Translation Working Group

Fatima Al-Shahrour[17], Gurnit Atwal[1,2,3], Peter J. Bailey[18], Andrew V. Biankin[19,20,21,22], Paul C. Boutros[1,23,24,25], Peter J. Campbell[26,27], David K. Chang[20,22], Susanna L. Cooke[22], Vikram Deshpande[28], Bishoy M. Faltas[29], William C. Faquin[28], Levi Garraway[30], Gad Getz[4,5,6,14], Sean M. Grimmond[31], Syed Haider[1], Katherine A. Hoadley[32,33], Wei Jiao[1], Vera B. Kaiser[34], Rosa Karlic[7], Mamoru Kato[35], Kirsten Kübler[4,5,28], Alexander J. Lazar[36], Constance H. Li[1,23], David N. Louis[28], Adam Margolin[37], Sancha Martin[26,38], Hardeep K. Nahal-Bose[39], G. Petur Nielsen[28], Serena Nik-Zainal[26,40,41,42], Larsson Omberg[43], Christine P'ng[1], Marc D. Perry[39,44], Paz Polak[4,5,6,16], Esther Rheinbay[4,5,28], Mark A. Rubin[45,46,47,48,49], Colin A. Semple[34], Dennis C. Sgroi[28], Tatsuhiro Shibata[50,51], Reiner Siebert[52,53], Jaclyn Smith[37], Lincoln D. Stein[1,2], Miranda D. Stobbe[54,55], Ren X. Sun[1], Kevin Thai[39], Derek W. Wright[56,57], Chin-Lee Wu[28], Ke Yuan[38,58,59] & Junjun Zhang[39]

[17]Bioinformatics Unit, Spanish National Cancer Research Centre (CNIO), Madrid 28029, Spain. [18]University of Glasgow, CRUK Beatson Institute for Cancer Research, Bearsden, Glasgow G61 1BD, UK. [19]South Western Sydney Clinical School, Faculty of Medicine, University of NSW, Liverpool, NSW 2170, Australia. [20]The Kinghorn Cancer Centre, Cancer Division, Garvan Institute of Medical Research, University of NSW, Sydney, NSW 2010, Australia. [21]West of Scotland Pancreatic Unit, Glasgow Royal Infirmary, Glasgow G31 2ER, UK. [22]Wolfson Wohl Cancer Research Centre, Institute of Cancer Sciences, University of Glasgow, Bearsden, Glasgow G61 1QH, UK. [23]Department of Medical Biophysics, University of Toronto, Toronto, ON M5S 1A8, Canada. [24]Department of Pharmacology, University of Toronto, Toronto, ON M5S 1A8, Canada. [25]University of California Los Angeles, Los Angeles, CA 90095, USA. [26]Wellcome Sanger Institute, Wellcome Genome Campus, Hinxton, Cambridge CB10 1SA, UK. [27]Department of Haematology, University of Cambridge, Cambridge CB2 2XY, UK. [28]Massachusetts General Hospital, Boston, MA 02114, USA. [29]Weill Cornell Medical College, New York, NY 10065, USA. [30]Dana-Farber Cancer Institute, Boston, MA 02215, USA. [31]University of Melbourne Centre for Cancer Research, The University of Melbourne, Melbourne, VIC 3052, Australia. [32]Department of Genetics, University of North Carolina at Chapel Hill, Chapel Hill, NC 27599, USA. [33]Lineberger Comprehensive Cancer Center, University of North Carolina at Chapel Hill, Chapel Hill, NC 27599, USA. [34]MRC Human Genetics Unit, MRC IGMM, University of Edinburgh, Edinburgh EH4 2XU, UK. [35]Department of Bioinformatics, Research Institute, National Cancer Center Japan, Tokyo 104-0045, Japan. [36]Departments of Pathology, Genomic Medicine, and Translational Molecular Pathology, The University of Texas MD Anderson Cancer Center, Houston, TX 77030, USA. [37]Oregon Health & Science University, Portland, OR 97239, USA. [38]University of Glasgow, Glasgow G61 1BD, UK. [39]Genome Informatics Program, Ontario Institute for Cancer Research, Toronto, ON M5G 0A3, Canada. [40]Academic Department of Medical Genetics, University of Cambridge, Addenbrooke's Hospital, Cambridge CB2 0QQ, UK. [41]MRC Cancer Unit, University of Cambridge, Cambridge CB2 0XZ, UK. [42]The University of Cambridge School of Clinical Medicine, Cambridge CB2 0SP, UK. [43]Sage Bionetworks, Seattle, WA 98109, USA. [44]Department of Radiation Oncology, University of California San Francisco, San Francisco, CA 94518, USA. [45]Bern Center for Precision Medicine, University Hospital of Bern, University of Bern, Bern 3008, Switzerland. [46]Department for Biomedical Research, University of Bern, Bern 3008, Switzerland. [47]Englander Institute for Precision Medicine, Weill Cornell Medicine and NewYork Presbyterian Hospital, New York, NY 10021, USA. [48]Meyer Cancer Center, Weill Cornell Medicine, New York, NY 10065, USA. [49]Pathology and Laboratory, Weill Cornell Medical College, New York, NY 10021, USA. [50]Division of Cancer Genomics, National Cancer Center Research Institute, Tokyo 104-0045, Japan. [51]Laboratory of Molecular Medicine, Human Genome Center, The Institute of Medical Science, The University of Tokyo, Minato-ku, Tokyo 108-8639, Japan. [52]Human Genetics, University of Kiel, Kiel 24118, Germany. [53]Institute of Human Genetics, Ulm University and Ulm University Medical Center, Ulm 89081, Germany. [54]CNAG-CRG, Centre for Genomic Regulation (CRG), Barcelona Institute of Science and Technology (BIST), Barcelona 08028, Spain. [55]Universitat Pompeu Fabra (UPF), Barcelona 08003, Spain. [56]MRC-University of Glasgow Centre for Virus Research, Glasgow G61 1QH, UK. [57]Wolfson Wohl Cancer Research Centre, Institute of Cancer Sciences, University of Glasgow, Bearsden G61 1QH, United Kingdom. [58]Cancer Research UK Cambridge Institute, University of Cambridge, Cambridge CB2 0RE, UK. [59]School of Computing Science, University of Glasgow, Glasgow G12 8RZ, UK

