## [Peer Review File · Nature Communications]

Editorial Note: During the second round of peer review, Reviewer #4 provided only comments to the editors and indicated that (s)he was satisfied with the author's revisions.

Reviewers' comments:

Reviewer #1 (Remarks to the Author):

In this manuscript the authors use the Pan-cancer Analysis of Whole Genomes 76 (PCAWG) data set, consisting of >2,800 primary tumors across 38 cancer types to develop classifiers. They use a random forest classifier using a leave 25% out strategy and data including SNV, CNV, INDEL, structural variant positions and the mutated genes and mutational pathway as models. They then merge the models and then perform classifications, with each cancer types having different combinations of models.

Major comments:

1. It is not clear if the method will outperform histopathological diagnostics by a pathologist.
2. The authors should compare their method with others such as gene expression based diagnostic classification.
3. The performance of the individual methods is modest for the majority of models (figure 1) except the mutation type and SNV position, and even for these the performance is modest with F1 score of <0.7 for the majority of the tumors. This would correspond in a failure to diagnose these cancers in 30% of cases which is high.
4. The merging of CLL with B-cell NHL and AML with myeloproliferative neoplasm is problematic as these are different diseases and not difficult to distinguish in clinical practice.
5. The merging of the models may be problematic given as it requires a prior knowledge of the diagnosis to choose specific models, and therefore would not perform well for cancers of unknown primaries.
6. The performance for the independent validation is sub optimal (figure 2), and also requires prior knowledge of the diagnosis to choose the models to combine.
7. Minimal novel knowledge about the biology or pathological processes for the tumors is described.

Reviewer #2 (Remarks to the Author):

In this manuscript Jiao et al present a machine learning classifier for the distinction of tumor types. The authors used whole genome sequencing data and extracted features based on the topological distribution of somatic passenger mutations, type of somatic mutations, and pathways that are altered in the tumour. The initial dataset consisted of 2606 tumours spanning 25 major types. Furthermore, an independent dataset of 1600 tumor was used to validate the classifier. However, for the validation only SNV and In/Del calls were available resulting in an incomplete usage of features (structural variants, copy number variants). The manuscript is well-presented, the figures are well drawn and the references are appropriate. This is a very interesting and topical study which a but still needs some minor revisions.

The accuracy of the system overall was 86%, with 14 of the 23 tumour types achieving recalls of 80% or higher leaving 40% of tumor types that did not achieve sufficient recalls. An obvious clinical application is the assignment of CUPS, however these contribute only to a small set of cancers. A much more important application would be the classification of molecular subtypes since these need differential therapeutic management. This is where the classifier fails. Although this is covered in the discussion it could be slightly extended to highlight the need of subtype classification.

It is not clearly described on which basis features were combined to evaluate the best match. Apparently the author did not combine the best single parameter features. Did they test all possible combinations?

It would be interesting to see how tumor fraction or DNA quality influences the classifier. Although

this information is certainly not available for these datasets, this is an issue that might to be discussed.

Moreover, I miss some technical data about depth of sequencing etc. A statement of costs/time, is something like this approach feasible at all in a clinical setting?

Table 3 does not seem informative and might be shifted to Supplementary data

Reviewer #3 (Remarks to the Author):

Summary:

This manuscript describes the application of machine learning to discern cancer type from whole genome sequencing derived somatic mutations in order to predict the site of origin of metastatic tumors when no primary can be found. This prediction problem is important because tumor origin is currently the best guideline for selecting effective therapies and 3-5% of cancer cases are of unknown primary origin. The authors find that the topographic distribution of somatic passenger mutations was the most predictive feature while mutation type and driver genes provide less information. This is a timely analysis that nicely ties together tumor type-specific genomic findings with performance of the machine learning classifier, providing insights into what is, and is not possible using DNA sequencing data. The manuscript is well written, the analysis presented is straightforward and the results are placed very well in the context of known tumor biology.

Major comments:

Since the goal of this approach is to enable classification of metastatic tumors to discern their origin, the authors should validate its performance on metastatic tumors. The authors removed all metastatic tumors from their dataset and never used them as validation. Although the authors find that performance is best when WGS is available, the WXS data available for the MET500 dataset of metastatic tumors may also serve as validation of performance when applied in the setting of metastasis. Although the performance when applied to WXS is decreased, if equivalent performance is obtained for primary versus metastatic samples it would show that the method should perform as well as it can in the setting for which it was designed.

The manuscript points to "Online Methods" in at least 2 places, however this reviewer did not find a formal methods section included in the package for review (Tables were included twice, so perhaps the incorrect file was uploaded). Prior to publication, the authors will need to provide a methods section with sufficient detail to allow reproduction of the analysis, including any parameters related to classifier construction such as the number of trees, the number of features evaluated each time a branch is added and feature selection criteria (Gini impurity, entropy, or something else?). Also, was the Random Forest score interpreted as a probability directly (Line 190), or was some other approach used to convert the score to a probability?

Relatedly, more detail should be provided about how features in the different categories were developed. Specifically, how were the gene and driver pathway features created? How was a mutation determined to 'impactful'? Although the performance of each feature category is provided, little information is provided about the relative importance of specific features in each category. The Random Forest algorithm can provide feature importances that can be used to evaluate the relative information provided by individual features.

Minor comments:

Since the authors are using a Random Forest classifier an unbiased estimate of the generalization

error can be obtained using out-of-bag samples, making cross-validation unnecessary. Was there any particular reason to prefer cross-validation? If so, it should be described somewhere.

The authors should cite and discuss the related work Soh, Kee Pang, et al. "Predicting cancer type from tumour DNA signatures." *Genome medicine* 9.1 (2017): 104. This method does not detract from the novelty of the current manuscript as it does not investigate distribution and signature of mutations as predictors and does not use a whole genome data set.

When there is ambiguity due to a driver gene/pathway being shared by different tumor types, it might be interesting to see if specific mutations can provide additional information about the primary. While including a large set of mutations might cause over-fitting, a small number of hotspot mutations with tumor-type specific biases might be informative.

For Figure 2, please include the number of validation samples for each tissue type.

The Figure 2 and 4 captions use "sensitivity" whereas the figures use "recall". Although most readers should know that the terms are synonymous, it would be best to be consistent.

Table 2 has a column "Feature Name" but the text uses the terminology "feature type".

There is a missing value on line 247.

Line 218: mutations used twice: "mutations in the NFE2L2 (NRF2) transcription factor mutations"

There is a typo in the last description of Table 2 "impactul".

Reviewer #4 (Remarks to the Author):

In this manuscript, the authors propose a machine learning classifier able to accurately distinguish 22 among 23 major cancer types using whole genome sequencing information. The classifier can achieve reasonably high accuracy when evaluated by cross validation and a separate patient cohort.

Our major question with this work is whether the authors actually solve the challenge they raise -- i.e., improved tissue identification from genome sequence. As the authors state, one of the reasons why solving this challenge might be necessary is that it is not always straightforward to determine where a metastatic tumour comes from. Therefore, for this work to be significant the authors need to show their model can achieve higher accuracy and/or provide value added compared to the information provided by current clinical practice, e.g. measurements from histology, radiology, and so on.

Here are several instances related to this problem in the paper:

1. In the Results section, the authors state "In some cases, the same donor had contributed both primary and metastatic tumour specimens to the PCAWG data set. In such cases, we used the primary tumour for training and evaluation". By excluding metastatic tumour samples in this study, how can its key questions be addressed? That is, how can we ever know whether the machine learning model actually determines the origin of a METASTATIC tumor, as proposed by the authors in the introduction?

2. Another appealing argument for why machine learning is necessary is the following: "Some other tumours are so poorly differentiated that they no longer express the cell-type specific proteins needed for unambiguous immunohistochemical classification". In the training data

however, due to poor prediction accuracy the authors merged chronic lymphocytic leukemia (CLL) and B-cell non-Hodgkins lymphoma (BNHL) into one tumour type, because they share a common trajectory of genomic alterations. Acute myeloid leukemia (AML) and myeloproliferative neoplasm (MPL) were also merged into one tumour type for the same reason. It seems that if two tumour samples do not express cell-type specific features then -- by definition based on how the training data are created -- the proposed machine learning model will not be able to distinguish them either.

3. In the "Patterns of Misclassification" section, the authors analyzed the reasons for misclassification of breast cancer. One of the reasons given is "breast cancer's highly heterogeneous molecular subtypes, among which is a basal subtype that shares molecular characteristics with high grade serous ovarian adenocarcinoma." This is likely also the reason why humans have a hard time determining tumour types. Where is the additional prediction power of machine learning in comparison to humans?

Another class of concerns relates to the machine learning model itself. Based on the authors' conclusions, the most powerful features they learned are individual gene-related features, such as KRAS, TP53, NFE2L2, PIK3R1, PTEN and PPP2R1A. If that is the case, why use a relatively complicated non-linear model? Non-linear models are designed to capture interactions between features; if genes are independent a linear model is more appropriate, powerful and generalizable. If the authors think there are some important interactions among genes in determination of tumour type, why are pathway features seldomly ranked as important features? Therefore, either the pathway features are wrongly designed or the non-linear model should be degenerated to a simple linear model to avoid overfitting.

Minor issues:

Across the cancer types, the performance of the machine learning model varies substantially. The authors studied several key factors which might influence this performance. We calculated the Pearson correlation between tumour sample sizes and prediction accuracy is reasonably high, $r = 0.42$. It thus seems that the model generally predicts better when there are more training samples. If the authors agree with this, maybe they should redo their analysis to make sure the high/low accuracy is indeed as they report, but not biased by sample size. If the authors do not agree that sample sizes influence performance, they should explain why they delete tumors with sample size smaller than 35 as they state in the paper.

The proposed machine learning model uses 14,202 features but is trained on fewer than 3000 tumours with multiple trees. The authors should state clearly how they control the model complexity to avoid overfitting.

We could not find the Online Methods.

Response to Reviewers

We thank the four reviewers for their thoughtful and helpful comments, and apologize for the long delay in submitting a revision. Since the original version, the manuscript has been substantially improved in the following areas:

- **Introduction of deep learning to improve classification accuracy.** The original version of this study used a random forest classifier to integrate multiple feature types. In the current version, we use a neural network to distinguish among tumour types, achieving a substantial increase in overall accuracy on both the internal and external validation sets. Overall accuracy for the task of distinguishing among 24 tumour types has risen from 86% to 91% on the internal PCAWG set, and from 80% to 85% and 82% on two independent validation sets.
- **Testing on metastatic tumours, including CUPs.** In response to reviewers' suggestions, we added an independent validation data set of over 2000 metastatic tumours, and show that classification accuracy on metastases is equivalent to its accuracy on independently sequenced primaries. The metastatic tumours include 62 cancers of unknown primary (CUPs) for which we provide tumour type predictions.
- **Slightly larger number of tumour types.** As suggested by two of the reviewers, we have unmerged the lymphoid and myeloid tumour types, increasing the number of tumour types to 24 without reducing overall accuracy.
- **Deeper discussion of other tumour classification tools,** including ones based on immunohistochemistry, mRNA, miRNA, DNA and cytosine methylation patterns. We also compare the accuracy of molecular classification tools to the accuracy of human experts using histopathology alone.
- **Demonstration that driver mutations make poor classifiers.** We explored the accuracy of classifiers trained with passenger mutations only, driver mutations only, or both. Surprisingly, classifiers trained just on altered driver genes and pathways perform poorly. In fact, the performance of classifiers trained on patterns of passenger mutations became slightly worse when driver-related features were added. We speculate that this reflects a combination of the sparsity of driver events combined with the universal "hallmarks of cancer" pathways shared in common among tumours. Note the move from a random forest classifier to one based on deep learning/neural networks made it more challenging to extract the features that contribute most strongly to the classifier. For this reason, the discussion of relative feature contribution has been curtailed in this revision.

In the remainder of this letter, we address each comment individually and provide a brief summary here. The reviewers' remarks are in blue, and our responses are in black.

Reviewer #1 (Remarks to the Author):

In this manuscript the authors use the Pan-cancer Analysis of Whole Genomes 76 (PCAWG) data set, consisting of >2,600 primary tumors across 38 cancer types to develop classifiers. They use a random forest

classifier using a leave 25% out strategy and data including SNV, CNV, INDEL, structural variant positions and the mutated genes and mutational pathway as models. They then merge the models and then perform classifications, with each cancer types having different combinations of models.

Major comments:

1. It is not clear if the method will outperform histopathological diagnostics by a pathologist.

There are few studies that directly test the accuracy of histopathological diagnosis of a metastasis when the pathologist is blinded to the patient's clinical history. However, we have found a 1993 study (Sheahan *et al.* Am. J. Clin. Pathol 1993;99:729-735) in which two pathologists were presented with 100 metastatic adenocarcinomas across 10 known primary sites and blinded to the identify of the primary. Based on histopathology alone, the pathologists chose the correct primary site less than half the time (49 and 47% accuracy for the senior and junior pathologist respectively). The correct primary site appeared among the pathologists' top three guesses 72 and 76% of the time. When minimal clinical information (age and sex) were provided, accuracy increased to 50 and 55%. Our classifier system compares favourably to this. Overall, we have an accuracy of 91% to distinguish among 24 different tumour types. Considering just the 11 adenocarcinoma tissue types among our data set, we have median recalls of 0.90 and 0.85 on the PCAWG primary and independent metastatic tumour data sets respectively.

A discussion of these points has been added to the paper.

2. The authors should compare their method with others such as gene expression based diagnostic classification.

We now include in the manuscript a discussion of the accuracy of multiple published tissue-of-origin methods using IHC, mRNA, miRNA and methylation. Reported accuracies range from 76% to 89% for assays that distinguish among 6 to 47 different tumour types, which is comparable to our system. However, an exhaustive comparison among the DNA-based method reported here and the expression- and methylation-based assays is difficult because of the different spectrum of tumour types reported by each assay, as well as the differing conditions each assay was designed for, for example fresh tissue versus formalin-fixed.

3. The performance of the individual methods is modest for the majority of models (figure 1) except the mutation type and SNV position, and even for these the performance is modest with F1 score of <0.7 for the majority of the tumors. This would correspond in a failure to diagnose these cancers in 30% of cases which is high.

The purpose for examining the performance of machine learning models on individual feature types was to understand their relative discriminatory power. The reviewer correctly points out that the majority of models based on single feature types have low accuracy. However, the following sections of the paper shows that classifier accuracy is substantially improved when multiple feature types are combined. Indeed, we achieve impressive levels of accuracy (median recall 0.91, precision 0.92, F1 0.90) with a deep learning neural network (DNN) that combines the two best-performing features types: the regional distribution of passenger mutations and the mutation type. Under ideal conditions in which WGS is performed using the same mutation calling methods that were applied to the training set, we expect to fail to classify tumours correctly less than 10% of the time.

A major takeaway from this section is that classifiers based on driver mutations does not perform well for many of the tumour types we examined. We have enhanced the text in this section to clarify the message.

4. The merging of CLL with B-cell NHL and AML with myeloproliferative neoplasm is problematic as these are different diseases and not difficult to distinguish in clinical practice.

Upon reflection, we felt that we could not justify the *ad hoc* merging of the two lymphoid and myeloid tumor types. In this version, we have redone the analysis without the prior merging of the two pairs of diagnoses. Unfortunately there were insufficient Myeloid-AML samples to train a classifier for this disease (just 11 donors), and so this tumour type was dropped. The remaining tumour types total 24, up from the original 23, and with the shift to a DNN classifier the overall accuracy actually improved. Perhaps not surprisingly, the DNN occasionally confused the two lymphoid tumour types. During internal validation, BNHL cases were incorrectly classified as CLL 5% of the time, and CLL cases were misclassified as BNHL 12% of the time. However, these were the only patterns of misclassification for the B-cell malignancies, and so the accuracy for predicting these tumour types remains good.

The text, figures and tables have all been modified accordingly.

5. The merging of the models may be problematic given as it requires a prior knowledge of the diagnosis to choose specific models, and therefore would not perform well for cancers of unknown primaries.

Agreed. We have eliminated the lymphoid and myeloid merging steps and trained classifiers on a total of 24 individual tumour types.

6. The performance for the independent validation is sub optimal (figure 2), and also requires prior knowledge of the diagnosis to choose the models to combine.

This appears to be a misunderstanding of the method for creating the combined classifiers. In the original manuscript, we trained each combined classifier using features from all the categories (SNV distribution, significantly mutated genes, structural variations) and then tested with held out sets for internal validation, and with an independent data set for external validation. After training of the models, no additional prior knowledge of the diagnosis was needed. The same applies to the current manuscript in which instead of applying multiple random forest binary classifiers to each tumour, we apply a single multi-class DNN that emits the probabilities that an unknown input tumour is a member of each of the tumour types.

We have rewritten this section for clarity. In addition, we have added an additional independent validation data set derived from WGS sequencing of ~2000 metastatic tumours. The performance of the DNN on this independent metastatic data set is the same as the performance on the independent primary tumour set, 82% overall..

7. Minimal novel knowledge about the biology or pathological processes for the tumors is described.

This is pretty much by design. This paper is intended to be part of a set of more than two dozen papers arising from the ICGC/TCGA Pan-Cancer Analysis of Whole Genomes (PCAWG) project which we hope to see published as a group in Nature Publishing Group journals. The biological insights gathered from this project are described in other papers that directly address patterns and mechanisms of structural variation, SNV distribution, and other genomic features. When appropriate, this manuscript does pick out biological findings from PCAWG that help explain the classifier's performance, most notably the connection between somatic mutation distribution and the chromatin state of the presumed cell of origin (see preprint at <https://doi.org/10.1101/517565>), the tumour type-specific nature of certain mutational signatures (see preprint at (<https://doi.org/10.1101/322859>) and the unexpected finding that passenger events are better discriminating features than driver events. Please see the marker paper preprint at: <https://www.biorxiv.org/content/early/2017/07/12/162784> for more information on the project and major scientific outputs.

In this revision, we have added references to other relevant PCAWG papers that are in various stages of review, to help readers identify studies of interest.

Reviewer #2 (Remarks to the Author):

In this manuscript Jiao et al present a machine learning classifier for the distinction of tumor types. The authors used whole genome sequencing data and extracted features based on the topological distribution of somatic passenger mutations, type of somatic mutations, and pathways that are altered in the tumour. The initial dataset consisted of 2606 tumours spanning 25 major types. Furthermore, an independent dataset of 1600 tumor was used to validate the classifier. However, for the validation only SNV and In/Del calls were available resulting in an incomplete usage of features (structural variants, copy number variants). The manuscript is well-presented, the figures are well drawn and the references are appropriate. This is a very interesting and topical study which a but still needs some minor revisions.

Thank you for the encouraging words!

The accuracy of the system overall was 86%, with 14 of the 23 tumour types achieving recalls of 80% or higher leaving 40% of tumor types that did not achieve sufficient recalls. An obvious clinical application is the assignment of CUPS, however these contribute only to a small set of cancers. A much more important application would be the classification of molecular subtypes since these need differential therapeutic management. This is where the classifier fails. Although this is covered in the discussion it could be slightly extended to highlight the need of subtype classification.

We strongly agree with this statement. Unfortunately due to the nature of the Pan-Cancer project we do not have sufficient numbers of specific tumour subtypes to adequately train the classifiers. As a result, what we have is a proof of principle that tumour subtyping using somatic mutation patterns should be possible given sufficiently-sized training sets. Encouragingly, we were able to show that the classifier is able to discriminate among a number of subtypes for which we had sufficient training sets. One example is renal cell carcinoma versus renal chromophobe carcinoma, and another is lung adenocarcinoma versus lung squamous cell carcinoma, We have added a discussion of these points to the manuscript.

It is not clearly described on which basis features were combined to evaluate the best match. Apparently the author did not combine the best single parameter features. Did they test all possible combinations?

The description of the feature selection steps in the original manuscript was inadequate. As a consequence of shifting to the DNN, the feature selection process changed significantly. There is now only a single multi-class classifier and it uses the same set of features (mutation type and distribution) for all tumour types. The process of hyperparameter selection, feature selection and training are described in detail in Online Methods.

It would be interesting to see how tumor fraction or DNA quality influences the classifier. Although this information is certainly not available for these datasets, this is an issue that might to be discussed.

We thank the reviewer for the suggestion to examine these factors. We did in fact test whether the classifier accuracy would be improved by looking exclusively at clonal mutations (which would be more reflective of early events during tumor initiation), but found no significant improvement during internal validation, and potentially a decrease in performance. Given that clonal and subclonal calls are not consistently distinguished in the external validation sets, we have not pursued the line of inquiry further.

Moreover, I miss some technical data about depth of sequencing etc. A statement of costs/time, is something like this approach feasible at all in a clinical setting?

The information on sequencing depth and other quality metrics is contained in the main PCAWG marker paper (preprint at <https://www.biorxiv.org/content/early/2017/07/12/162784>). We have copied the key QC metrics for the data set into the Results section of the manuscript.

The question about cost and time is a good one. The short answer is that the cost of performing WGS on a tumor/normal pair continues to fall and is currently in the \$3000-4000 USD range and 2-3 weeks for sequencing and full analysis. The more thoughtful answer is that there's a lot of additional clinically useful information to be gained from genome sequencing with respect to finding actionable mutations, measuring neo-antigen load, and identifying predictive/prognostic biomarkers. We are now seeing genome sequencing being applied clinically to routine cancer care, most spectacularly in the UK where the National Health Service recently announced a plan to apply WGS routinely to cancer patients (The Guardian, 3 July 2018, <https://goo.gl/TDx81P>). Since cancer genomes are going to be sequenced anyway in the clinical context,

the incremental cost of adding tumour type/subtype determination to the downstream analysis is low. We have added a brief discussion of this point.

Table 3 does not seem informative and might be shifted to Supplementary data

We have moved Table 3 into the supplement.

Reviewer #3 (Remarks to the Author):

Summary:

This manuscript describes the application of machine learning to discern cancer type from whole genome sequencing derived somatic mutations in order to predict the site of origin of metastatic tumors when no primary can be found. This prediction problem is important because tumor origin is currently the best guideline for selecting effective therapies and 3-5% of cancer cases are of unknown primary origin. The authors find that the topographic distribution of somatic passenger mutations was the most predictive feature while mutation type and driver genes provide less information. This is a timely analysis that nicely ties together tumor type-specific genomic findings with performance of the machine learning classifier, providing insights into what is, and is not possible using DNA sequencing data. The manuscript is well written, the analysis presented is straightforward and the results are placed very well in the context of known tumor biology.

Thank you for the kind words!

Major comments:

Since the goal of this approach is to enable classification of metastatic tumors to discern their origin, the authors should validate its performance on metastatic tumors. The authors removed all metastatic tumors from their dataset and never used them as validation. Although the authors find that performance is best when WGS is available, the WXS data available for the MET500 dataset of metastatic tumors may also serve as validation of performance when applied in the setting of metastasis. Although the performance when applied to WXS is decreased, if equivalent performance is obtained for primary versus metastatic samples it would show that the method should perform as well as it can in the setting for which it was designed.

This is an extremely good point and we are embarrassed that we missed the opportunity to explore this question in the original manuscript. We have now extended the validation to a set of 2,120 WGS on metastatic tumours with known primaries released in pre-publication form by the Hartwig Medical Foundation (BioRxiv preprint DOI <https://doi.org/10.1101/415133>), and used with permission of the authors. We supplemented this with a set of 96 metastatic pancreatic adenocarcinoma WGS from the COMPASS study (Clinical Cancer Research, 24(6), 1344–1354. (2018). Gratifyingly, the DNN is just as accurate at predicting the primary from metastatic tumour as it is for the independent primary tumour validation set, and we have added a new section that describes these results. To compensate for the increase in word length, we have removed the section in which we tested the classifier against simulated whole exomes.

The manuscript points to “Online Methods” in at least 2 places, however this reviewer did not find a formal methods section included in the package for review (Tables were included twice, so perhaps the incorrect file was uploaded). Prior to publication, the authors will need to provide a methods section with sufficient detail to allow reproduction of the analysis, including any parameters related to classifier construction such as the number of trees, the number of features evaluated each time a branch is added and feature selection criteria (Gini impurity, entropy, or something else?). Also, was the Random Forest score interpreted as a probability directly (Line 190), or was some other approach used to convert the score to a probability?

We apologize for the error. Indeed the wrong file appears to have been uploaded during the original submission. The current submission contains the correct Online Methods file, and it addresses the methodological questions that the reviewer has raised.

Relatedly, more detail should be provided about how features in the different categories were developed. Specifically, how were the gene and driver pathway features created? How was a mutation determined to ‘impactful’? Although the performance of each feature category is provided, little information is provided about the relative importance of specific features in each category. The Random Forest algorithm can provide feature importances that can be used to evaluate the relative information provided by individual features.

The requested details are described in the now-available Methods section. Briefly, significantly mutated genes were determined in a separate effort by the PCAWG Drivers and Functional Interpretation Working

Group (described in <https://www.biorxiv.org/content/early/2017/12/23/237313>), using a consensus method in which coding and noncoding genes, cis-regulatory regions, and other functional elements were examined for a statistical excess of somatic mutations and/or impactful mutations. "Impact" is defined differently for each type of element: in the case of coding genes the impact score takes into account whether the mutation causes a nonsynonymous amino acid change and if so whether the residue occurs in a conserved protein domain. In the case of cis-regulatory region mutations, the impact score reflects whether a known transcriptional factor binding site motif is altered. For pathway features, we tallied all protein-coding genes containing non-synonymous SNVs, and assigned to 1,865 pathways from the Reactome resource (<http://www.reactome.org>, version 58). A pathway feature was scored as positive if it contained at least one driver gene.

With respect to deriving relative feature importance directly from the merged feature Random Forest trees, this comment is now moot because we shifted from running 24 random forest binary classifiers to a single multi-class neural network. However, we still use Random Forests to build and compare classifiers from each of the single feature types (Figure 1 and Supplementary Figure 1), and find that mutation type and regional distribution are the most consistent discriminative feature. Surprisingly, driver related features (mutated driver genes and pathways) were usually among the worst performing features.

Minor comments:

Since the authors are using a Random Forest classifier an unbiased estimate of the generalization error can be obtained using out-of-bag samples, making cross-validation unnecessary. Was there any particular reason to prefer cross-validation? If so, it should be described somewhere.

This comment is now moot because of the shift from a Random Forest classifier to a neural net for the main classifier. The process of training and internally validating the neural net is described in the Online Methods section.

The authors should cite and discuss the related work Soh, Kee Pang, et al. "Predicting cancer type from tumour DNA signatures." *Genome medicine* 9.1 (2017): 104. This method does not detract from the novelty of the current manuscript as it does not investigate distribution and signature of mutations as predictors and does not use a whole genome data set.

Indeed, this paper came to our attention only after we had submitted the first version of the manuscript. The revision cites and discusses this work and other recent related papers.

When there is ambiguity due to a driver gene/pathway being shared by different tumor types, it might be interesting to see if specific mutations can provide additional information about the primary. While including a large set of mutations might cause over-fitting, a small number of hotspot mutations with tumor-type specific biases might be informative.

We appreciate this suggestion and have experimented with various ways of implementing it. Unfortunately we haven't yet demonstrated an appreciable improvement over the current, approach. We continue to experiment with ways of improving classifier accuracy. In addition to the approach suggested by the reviewer, we have tried looking at clonal vs subclonal mutations separately, adding features for tumor purity and ploidy, and using dynamically-sized bins for the passenger mutation distribution. However, none of the refinements we've tried have consistently improved the performance of the classifier. A simple approach using just regional passenger SNV density and the mutation type profile seems to be adequate to distinguish the tumour types in our collection.

For Figure 2, please include the number of validation samples for each tissue type.

Thank you for the great suggestion. This has been done. The PCAWG sample set sizes are now indicated in Table 1 and Figures 2, and the size of the primary and metastatic tumour independent validation sets are indicated in Supplementary Table 4 and Figure 4.

The Figure 2 and 4 captions use "sensitivity" whereas the figures use "recall". Although most readers should know that the terms are synonymous, it would be best to be consistent.

Fixed.

Table 2 has a column "Feature Name" but the text uses the terminology "feature type".

Fixed.

There is a missing value on line 247.

Fixed.

Line 218: mutations used twice: "mutations in the NFE2L2 (NRF2) transcription factor mutations"

Fixed.

There is a typo in the last description of Table 2 "impactul".

Fixed.

Reviewer #4 (Remarks to the Author):

In this manuscript, the authors propose a machine learning classifier able to accurately distinguish 22 among 23 major cancer types using whole genome sequencing information. The classifier can achieve reasonably high accuracy when evaluated by cross validation and a separate patient cohort.

Our major question with this work is whether the authors actually solve the challenge they raise -- i.e., improved tissue identification from genome sequence. As the authors state, one of the reasons why solving this challenge might be necessary is that it is not always straightforward to determine where a metastatic tumour comes from. Therefore, for this work to be significant the authors need to show their model can achieve higher accuracy and/or provide value added compared to the information provided by current clinical practice, e.g. measurements from histology, radiology, and so on.

We've taken this set of comments very much to heart and in response have improved the study in several ways described in the responses below.

Here are several instances related to this problem in the paper:

1. In the Results section, the authors state "In some cases, the same donor had contributed both primary and metastatic tumour specimens to the PCAWG data set. In such cases, we used the primary tumour for training and evaluation". By excluding metastatic tumour samples in this study, how can its key questions be addressed? That is, how can we ever know whether the machine learning model actually determines the origin of a METASTATIC tumor, as proposed by the authors in the introduction?

The reviewer has put his or her finger on a key weakness in the original study. Fortunately we've been able to address the question at least partially by extending the study to metastatic tumors. We have now extended the validation to a set of 2,120 WGS on metastatic tumours with known primaries released in pre-publication form by the Hartwig Medical Foundation (BioRxiv preprint DOI <https://doi.org/10.1101/415133>), and used with permission of the authors. We supplemented this with a set of 96 metastatic pancreatic adenocarcinoma WGS from the COMPASS study (Clinical Cancer Research, 24(6), 1344–1354. (2018). Gratifyingly, the DNN is just as accurate at predicting the primary from metastatic tumour as it is for the independent primary tumour validation set, and we have added a new section that describes these results. To compensate for the increase in word length, we have removed the section in which we tested the classifier against simulated whole exomes.

2. Another appealing argument for why machine learning is necessary is the following: "Some other tumours are so poorly differentiated that they no longer express the cell-type specific proteins needed for unambiguous immunohistochemical classification". In the training data however, due to poor prediction accuracy the authors merged chronic lymphocytic leukemia (CLL) and B-cell non-Hodgkins lymphoma (BNHL) into one tumour type, because they share a common trajectory of genomic alterations. Acute myeloid leukemia (AML) and myeloproliferative neoplasm (MPL) were also merged into one tumour type for the same reason. It seems that if two tumour samples do not express cell-type specific features then -- by definition based on how the training data are created -- the proposed machine learning model will not be able to distinguish them either.

Thanks for this comment. Upon reflection, we felt that we could not justify the *ad hoc* merging of the two lymphoid and myeloid tumor types. In this version, we have redone the analysis without the prior merging of the two pairs of diagnoses. Unfortunately there were insufficient Myeloid-AML samples to train a classifier for this disease (just 11 donors), and so this tumour type was dropped. The remaining tumour types total 24, up from the original 23, and with the shift to a DNN classifier the overall accuracy actually improved. Perhaps not surprisingly, the DNN occasionally confused the two lymphoid tumour types. During internal validation, BNHL cases were incorrectly classified as CLL 5% of the time, and CLL cases were misclassified as BNHL 12% of the time. However, these were the only patterns of misclassification for the B-cell malignancies, and so the accuracy for predicting these tumour types remains good.

The text, figures and tables have all been modified accordingly.

Nevertheless, the reviewer's essential point still stands. If two tumors are biologically similar, they cannot easily be distinguished. This may explain the frequent confusion of esophageal and gastric cancer, and is the explanation we offer for this observation in the patterns of misclassification section.

It is also worth noting that we were making a slightly different point in the sentence quoted by the reviewer. The idea we tried to express is that the majority of somatic mutations arise in the cell of origin prior to its malignant transformation, and are therefore indelible marks of the differentiation state of the cell of origin at the time the tumor arose. This is in distinction to the expression state of the mature tumor, which could, in theory, undergo de-differentiation or trans-differentiation, losing diagnostic protein and/or expression patterns. This hypothesis isn't necessarily inconsistent with the difficulty in distinguishing esophageal and gastric cancer, or confusing lung and head/neck squamous cell cancer. It simply means that the cells of origin of these tumor types are either the same or very similar to each other.

3. In the "Patterns of Misclassification" section, the authors analyzed the reasons for misclassification of breast cancer. One of the reasons given is "breast cancer's highly heterogeneous molecular subtypes, among which is a basal subtype that shares molecular characteristics with high grade serous ovarian adenocarcinoma." This is likely also the reason why humans have a hard time determining tumour types. Where is the additional prediction power of machine learning in comparison to humans?

The reviewer's insight reflects a fundamental problem with supervised machine learning: if the labels applied to the samples do not reflect an underlying truth, then the classifier will perform poorly. Our current way of classifying cancers is based on an anatomic and histological taxonomy that goes back to the 19th century and is certainly in need of revision. Breast cancer is a great example of this. Before the advent of molecular techniques, breast cancer subtypes were defined entirely by histological criteria such as the presence of recognizable ductal architecture. We now know that the histological subtypes obscure fundamental biological differences such as the presence of HER2 driver amplifications and the expression of hormone receptors. There's also good evidence that at least some breast cancer subtypes are related to the particular normal cell of origin from which the malignant cell derives. In general, the human eye cannot distinguish these molecular subtypes, at least not using routine pathological stains. Since the discovery of molecular subtypes of breast cancer, tremendous knowledge has been gained on the different prognosis and therapeutic responses of the subtypes. For this reason, the standard of care for breast cancer now involves performing molecular subtyping so that the patient can be given the course of treatment that is most likely to produce a response.

One of the basic goals of cancer genomics is to use **unsupervised** learning to discover latent subtypes among histologically-similar tumors, as well as to discover the biological underpinnings of classically-defined subtypes. For example, the beautiful recent TCGA Cell paper *Cell-of-Origin Patterns Dominate the Molecular Classification of 10,000 Tumors from 33 Types of Cancer* (doi: <https://doi.org/10.1016/j.cell.2018.03.022>) used unsupervised clustering across diverse molecular features to identify multiple types and subtypes of cancers; the clusters obtained overlap with the classical types via a complex set of merges and splits. In principle, we could discard the classical histopathological labels and retrain our supervised classifier using the discovered molecular types, and we suspect our overall accuracy would be improved. In practice, however, it would be premature to do this, as the molecular subtypes discovered in this way haven't yet been validated and don't have diagnostic or therapeutic knowledge attached to them. It would also be practically difficult to work with the TCGA tumors, which do not have whole genome sequencing data available.

Ironically, when we shifted from a series of Random Forest classifiers to a single multi-class neural network, the issues with misclassification errors among breast cancer cases disappeared, and breast adenocarcinomas are now among the top five most accurate classes (F1 0.93 on internal validation; 0.98 on external validation set). The statements that triggered the reviewer's comments are no longer included in the manuscript.

Another class of concerns relates to the machine learning model itself. Based on the authors' conclusions, the most powerful features they learned are individual gene-related features, such as KRAS, TP53, NFE2L2, PIK3R1, PTEN and PPP2R1A. If that is the case, why use a relatively complicated non-linear model? Non-linear models are designed to capture interactions between features; if genes are independent a linear model is more appropriate, powerful and generalizable. If the authors think there are some important interactions among genes in determination of tumour type, why are pathway features seldomly ranked as

important features? Therefore, either the pathway features are wrongly designed or the non-linear model should be degenerated to a simple linear model to avoid overfitting.

The reviewer seems to have misinterpreted our results. Under *Tumour-Specific Features* we stated:

The majority of classifiers selected mutation type and/or mutation distribution as the most influential features, emphasizing the importance of exposures and epigenetically-related cell-of-origin marks in distinguishing tumour types....In some cases the classifiers identified individual gene-related features and focal variants that distinguish one tumour type from others

In fact, gene-related features were useful in only six of the 24 tumour types. For all the others, mutation signatures and/or mutation distribution were selected by the classifiers.

In the current manuscript, we shifted from Random Forest models to a Neural Network for tumour type classification based on combinations of feature types. Rather surprisingly, we found that just the passenger SNV regional distribution, and the passenger SNV type frequencies were sufficient for accurate classification. In fact, when we added information on altered driver genes and/or driver pathways, classifier accuracy became slightly worse. When only driver and pathway information is available, the accuracy became much worse -- accuracy drops from 91% to 40%. We think that this reflects the relatively low number of identified driver events (4.6 per patient in the training cohort), combined with sharing of similar pathway alterations reflecting the common "hallmarks of cancer" thought to unite all tumour types. We've added a discussion of this finding to the current revision.

Minor issues:

Across the cancer types, the performance of the machine learning model varies substantially. The authors studied several key factors which might influence this performance. We calculated the Pearson correlation between tumour sample sizes and prediction accuracy is reasonably high, $r = 0.42$. It thus seems that the model generally predicts better when there are more training samples. If the authors agree with this, maybe they should redo their analysis to make sure the high/low accuracy is indeed as they report, but not biased by sample size. If the authors do not agree that sample sizes influence performance, they should explain why they delete tumors with sample size smaller than 35 as they state in the paper.

This is an extremely good point, and we are chagrined that we didn't highlight it more prominently in the original manuscript. We thank the reviewer for the suggestion to present the correlation between accuracy and sample size, have added an explicit discussion of training set sizes to the section *Classification using Combinations of Mutation Feature Types*, and have illustrated the effect in a new figure panel, Figure 3A.

The proposed machine learning model uses 14,202 features but is trained on fewer than 3000 tumours with multiple trees. The authors should state clearly how they control the model complexity to avoid overfitting.

Since the machine learning model has changed, this comment no longer directly applies. We discuss the methodology for selecting hyperparameters and training the Neural Network in Online Methods. Briefly, our control for overfitting is to partition the sample set into independent training and validation sets ten times, build the classifier from the training set partition, test it against the validation partition, and then report the average of the ten training/validation cycles. To test against the external validation sets (one each for primary and metastatic tumours), we created an ensemble model that averaged the output of the ten independently trained models, and applied it directly to passenger SNV data derived from the two sets without performing any further training.

We could not find the Online Methods.

We apologize for the omission. We appear to have uploaded the incorrect PDF file during the original submission. The current submission contains the correct Methods file.

REVIEWERS' COMMENTS:

Reviewer #1 (Remarks to the Author):

In this revised manuscript the authors use the Pan-cancer Analysis of Whole Genomes (PCAWG) data set, now consisting of 2606 tumors across 24 cancer types to develop classifiers. They have now added deep learning neural network (DNN) and found that using the regional distribution of passenger mutations and the mutation type produces significantly improved accuracy (median recall 0.91, precision 0.92, F1 0.90). They have also responded to the majority critiques, including adding metastatic tumors.

Minor comments:

1. The references are not correctly cited and there appears to be several that point to the wrong citations at the end.
2. Many of the figure fonts are small and difficult to read and should be enlarged.
3. Figure 3B needs to be better described in the legend as it is hard to interpret.
4. Discussion on the cost and speed of sequencing can be much reduced as it will rapidly change.

Reviewer #2 (Remarks to the Author):

This a substantially improved version of the manuscript. In addition to the RF model, the authors employed DNN to distinguish among tumour types and achieved an overall accuracy of 91% for the PCAWG set. Although some tumour types performed very well, others still caused difficulties for classification. However, this has been acknowledged in the manuscript. In addition, the authors used independent cohort for validation which included 2000 metastatic tumours and 62 CUPs. The CUP section is rather short, however, apparently the authors did not have ground truth for these tumours. Also, feature selection is now adequately described and the authors put their classifier into context with other tumour classification tools. Taken together, the authors addressed most of the reviewer's concerns.

Minor comments:

Figure 2: for some types the sum of the classified tumours exceeds 100%, e.g. Breast cancer 107%. Please explain how explain how the figures were rounded.
For Supplementary figures headings and legends are missing.

Reviewer #3 (Remarks to the Author):

The authors have done significant work to address reviewer comments.

I had one additional minor comment:

In the abstract: "but in 3% of the time a cancer patient presents with metastatic tumour and no obvious primary" should either be "3% of the time" or "in 3% of cancer patients" – I am not sure which variant is better, but "in 3% of the time" is grammatically incorrect.

REVIEWERS' COMMENTS:

Reviewer #1 (Remarks to the Author):

In this revised manuscript the authors use the Pan-cancer Analysis of Whole Genomes (PCAWG) data set, now consisting of 2606 tumors across 24 cancer types to develop classifiers. They have now added deep learning neural network (DNN) and found that using the regional distribution of passenger mutations and the mutation type produces significantly improved accuracy (median recall 0.91, precision 0.92, F1 0.90). They have also responded to the majority critiques, including adding metastatic tumors.

Thank you! We did our best to address your comments.

Minor comments:

1. The references are not correctly cited and there appears to be several that point to the wrong citations at the end.

We think we have found and corrected all mis-citations.

2. Many of the figure fonts are small and difficult to read and should be enlarged.

We have enlarged the figure fonts in the high-resolution images.

3. Figure 3B needs to be better described in the legend as it is hard to interpret.

We have re-worded the legend. The new wording is:

(B) Accuracy of the classifier when it is asked to identify the correct tumor type among its top N ranked predictions. The blue dashed line is the median true positive rate among all 24 tumor classes. The green and red dashed lines correspond to the true positive rate for the best and worst-performing tumor classes.

4. Discussion on the cost and speed of sequencing can be much reduced as it will rapidly change.

Specific numbers on the cost and speed of sequencing have been deleted and have been replaced with a more generic (and hopefully uncontroversial statement):

In practical terms, whole genome sequencing and analysis of cancers is becoming increasingly cost effective, and there is an accelerating trend to apply genome sequencing to routine cancer care in order to identify actionable mutations and to test for the presence of predictive biomarkers.

Reviewer #2 (Remarks to the Author):

This a substantially improved version of the manuscript. In addition to the RF model, the authors employed DNN to distinguish among tumour types and achieved an overall accuracy of 91% for the PCAWG set. Although some tumour types performed very well, others still caused difficulties for classification. However, this has been acknowledged in the manuscript. In addition, the authors used

independent cohort for validation which included 2000 metastatic tumours and 62 CUPs. The CUP section is rather short, however, apparently the authors did not have ground truth for these tumours. Also, feature selection is now adequately described and the authors put their classifier into context with other tumour classification tools. Taken together, the authors addressed most of the reviewer's concerns.

We thank the reviewer for their kind remarks.

The reviewer is quite right that we do not have the ground truth for the CUP tumors; this is the very definition of CUPs! Indeed, we struggled to decide how to present these results. We hoped at first to compare the rates at which the classifier assigned various tumor types to the CUP cases to published studies of tumors that were initially called CUPs and then assigned origins using IHC or molecular studies. However, the rates in these studies vary considerably due to methodological and ascertainment issues, and we settled in the end for a simple enumeration of the classifier predictions.

Minor comments:

Figure 2: for some types the sum of the classified tumours exceeds 100%, e.g. Breast cancer 107%. Please explain how explain how the figures were rounded.

This is simply from rounding errors. For example, breast cancer has multiple cells with values between 0.5 and 0.9% and these were rounded up to integers in order for the percentages to fit inside the cell space. We have added a short explanation to the figure legend.

For Supplementary figures headings and legends are missing.

We are sorry. The Supplementary headings and legends were in a separate document that may not have been submitted properly. We have now moved the Supplementary headings and legends into the main manuscript.

Reviewer #3 (Remarks to the Author):

The authors have done significant work to address reviewer comments.

Thanks so much!

I had one additional minor comment:

In the abstract: "but in 3% of the time a cancer patient presents with metastatic tumour and no obvious primary" should either be "3% of the time" or "in 3% of cancer patients" – I am not sure which variant is better, but "in 3% of the time" is grammatically incorrect.

We have changed this to "but in 3% of cases a patient presents with metastatic tumour and no obvious primary".